# Towards Continuous Reuse of Graph Models via Holistic Memory Diversification

**Ziyue Qiao**[1]  **Junren Xiao**[2]  **Qingqiang Sun**[3,*]  **Meng Xiao**[4]  **Xiao Luo**[5]  **Hui Xiong**[2,*]

[1]School of Computing and Information Technology, Great Bay University
[2]AI Thrust, The Hong Kong University of Science and Technology (Guangzhou)
[3]School of Engineering, Great Bay University
[4]Computer Network Information Center, Chinese Academy of Sciences
[5]Department of Computer Science, University of California, Los Angeles
`{ziyuejoe, junrenxiao138, sunisfighting}@gmail.com`
`shaow@cnic.com, xiaoluo@cs.ucla.edu, xionghui@ust.hk`

## ABSTRACT

This paper addresses the challenge of incremental learning in growing graphs with increasingly complex tasks. The goal is to continuously train a graph model to handle new tasks while retaining proficiency on previous tasks via memory replay. Existing methods usually overlook the importance of memory diversity, limiting in selecting high-quality memory from previous tasks and remembering broad previous knowledge within the scarce memory on graphs. To address that, we introduce a novel holistic Diversified Memory Selection and Generation (DMSG) framework for incremental learning in graphs, which first introduces a buffer selection strategy that considers both intra-class and inter-class diversities, employing an efficient greedy algorithm for sampling representative training nodes from graphs into memory buffers after learning each new task. Then, to adequately rememorize the knowledge preserved in the memory buffer when learning new tasks, a diversified memory generation replay method is introduced. This method utilizes a variational layer to generate the distribution of buffer node embeddings and sample synthesized ones for replaying. Furthermore, an adversarial variational embedding learning method and a reconstruction-based decoder are proposed to maintain the integrity and consolidate the generalization of the synthesized node embeddings, respectively. Extensive experimental results on publicly accessible datasets demonstrate the superiority of DMSG over state-of-the-art methods. The code and data can be found in `https://github.com/joe817/DMSG`.

## 1 INTRODUCTION

Graphs, owing to their flexible relational data structures, are widely employed for many applications in various domains, including social networks (Jiang et al., 2016; Qiao et al., 2021), recommendation systems (Wu et al., 2022), and bioinformatics (Zhang et al., 2021; Huang et al., 2025). With the increasing prevalence of graph data, graph-based models like Graph Neural Networks (GNNs) have gained significant attention due to their ability to capture complex structural relationships and those dynamic variants also demonstrate remarkable inductive capabilities on growing graph data (Ju et al., 2024; Pareja et al., 2020; Manessi et al., 2020). However, as the growing graph in Figure 1 shows, when new nodes are added, the associated learning tasks can become increasingly complex. For example, the graph models on academic networks might need to predict the topics of papers in highly dynamic research areas where the topic rapidly emerges, and those on recommendation networks might need to continually adapt to new user preferences. Recent research on the inductive capability and adaptability of GNNs often remains limited to a specific task (Hamilton et al., 2017; Han et al., 2021; Qiao et al., 2023) and cannot be readily applied to incremental tasks. Moreover, it is often inefficient to train an entirely new model from scratch every time a new learning task is introduced. In recent years, model reuse via incremental learning (Tan et al., 2022; Kim et al., 2022), also known

---
*Corresponding Authors

as continual learning or lifelong learning, has led to exploring more economically viable pipelines, enabling the model to adaptively learn new tasks while maintaining the knowledge from old tasks.

The main challenge of incremental learning on graphs lies in mitigating catastrophic forgetting. As the graph model learns from a sequence of tasks on evolving graphs, it tends to forget the information learned from previous tasks when acquiring knowledge from new tasks. One prevalent approach to address this issue is the memory replay method, a human-like method that typically maintains a memory buffer to store the knowledge gained from previous tasks. When learning a new task, the model not only focuses on the current information but also retrieves and re-learns from memory, preventing the model from forgetting what was learned previously as it takes on new tasks. Nevertheless, this method has two major focuses to address on graphs: (1) **How to select knowledge from old graphs to form more high-quality memory buffers?** Existing methods usually select representative training samples as knowledge. However, determining which nodes in the graph are more representative is difficult and usually a time-consuming process. Furthermore, most methods (Zhang et al., 2022c; Wang et al., 2022) select samples all into one same buffer without considering the inter-class differences between various previous tasks, which may degrade the quality of preserved knowledge. (2) **How to effectively replay the limited buffer knowledge to enhance the model's memorization of previous tasks?** Due to constraints related to memory and training expenses, the samples chosen for the buffer are often limited. Many methods (Zhou & Cao, 2021; Su & Wu, 2023) concentrate on memory selection, neglecting to broaden the boundaries of memory within the buffer, resulting in a discount in replay. Finding an effective way to replay knowledge from these limited nodes is critical to incremental learning in graphs.

In this paper, we propose a novel Diversified Memory Selection and Generation (DMSG) method on incremental learning in growing graphs, devised to tackle the above challenges. we consider that selecting diversified memory helps in **Comprehensive Knowledge Retention**: we apply a heuristic diversified memory selection strategy that takes into account both intra-class and inter-class diversities between nodes. By employing an efficient greedy algorithm, we selectively sample representative training nodes from the growing graph, placing them into memory buffers after completing each new learning task. Furthermore, we explore the memory diversification in memory reply for **Enhanced Knowledge Memorization**: we introduce a generative memory replay method, which first leverages a variational layer to produce the distribution of buffer node embeddings, from which synthesized samples are drawn for replaying. We incorporate an adversarial variational embedding learning technique and a reconstruction-based decoder. These are designed to preserve the integrity and strengthen the generalization of the synthesized node embeddings on the label space, ensuring the essential knowledge is carried over accurately and effectively.

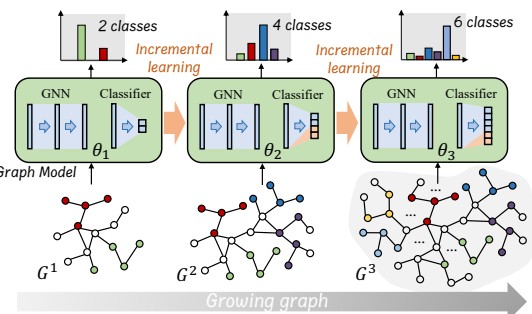

Figure 1: An example of incremental learning in growing graphs, where nodes with distinct labels are shaded in various colors. The number of classes expands as the graph grows, causing increasingly complex classification tasks.

The main contributions can be summarized as follows: (1) We propose a novel and effective memory buffer selection strategy that considers both the intra-class and inter-class diversities to select representative nodes into buffers. (2) We propose a novel memory replay generation method on graphs to generate diversified and high-quality nodes from the limited real nodes in buffers, exploring the essential knowledge and enhancing the effectiveness of replaying. (3) Extensive experiments on various incremental learning benchmark graphs demonstrate the superiority of the proposed DMSG over state-of-the-art methods.

## 2 PROBLEM FORMULATION

In this section, we present the formulation for the incremental learning problem in growing graphs. Generally, a growing graph is represented by a sequential of $m$ snapshots: $G = \{G^1, G^2, ..., G^m\}$,

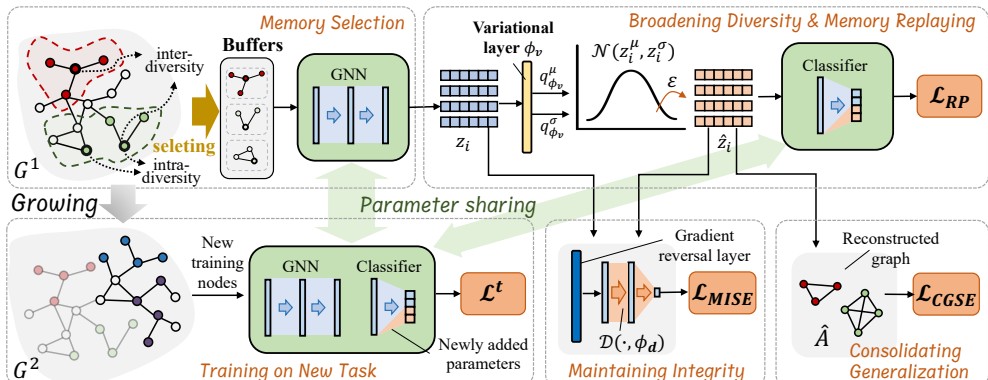

Figure 2: The framework of DMSG. In this instance, the graph model underwent training on a 2-class node classification task on $G^1$. Two new classes of nodes are added to $G^1$ to form $G^2$. Certain nodes of the previous two classes are first selected into buffers. Then, the model is further trained on the two new classes of nodes and buffers to perform incremental learning.

and each snapshot corresponds to the inception of a new task, represented as $\mathcal{T} = \{\mathcal{T}^1, \mathcal{T}^2, ..., \mathcal{T}^m\}$. Each graph $G^i$ is evolved from the previous graph $G^{i-1}$, i.e., $G^{i-1} \subset G^i, \forall i \in 2, ..., m.$, and each learning task $\mathcal{T}^i$ is more complex than the previous task $\mathcal{T}_{i-1}$. This paper specifies the learning tasks to classification tasks and followss the class-incremental learning setting, i.e., the number of classes increases alongside graph growth, increasing the task complexity. In this scenario, we aim to continuously learn a model $f(\theta)$ on $\mathcal{T}$. For the $t$-th step, the task $\mathcal{T}^t$ incorporates a training node set $\mathcal{V}^t$ with previously unseen labels (i.e., novel classes), where each vertex $v_i \in \mathcal{V}^t$ has the label $y_i \in \mathcal{Y}^t$, and $\mathcal{Y}^t = \{y_1^t, y_2^t, ..., y_n^t\}$ is the set of $n$ novel classes. The task $\mathcal{T}^t$ is to train the $f(\theta) : v_i \rightarrow \bigcup_{j=1}^t \mathcal{Y}^j$ to ensure it can infer well on the current novel classes while preventing catastrophic forgetting of the inference ability on previous classes.

**Jointly Incremental Learning.** This is a straightforward solution for the problem, which collects training nodes of all classes of previous tasks to train $f(\theta)$ in each step. This treats the accumulated tasks $\{\mathcal{T}^1, \mathcal{T}^2, ..., \mathcal{T}^t\}$ as a whole new task and retrains the model from scratch. However, this solution is inefficient because it leads to redundant training of labeled nodes and creates computational challenges due to the growing graph size.

**Memory Replay for Incremental Learning.** This method offers a more practical solution. Instead of gathering all previous training nodes, it maintains buffers $\mathcal{B}$ that store a small yet representative subset of training nodes for each class of previous tasks. The objectives can be formulated as follows:

$$\mathcal{L} = \underbrace{\sum_{v_i \in \mathcal{V}^t} \ell^t(f(v_i; \theta), y_i)}_{\text{Loss on new tasks}} + \lambda \underbrace{\sum_{j=1}^{K} \sum_{v_k \in \mathcal{B}_j} \ell^t(f(v_k; \theta), y_k)}_{\text{Memory replay on previous tasks}}, \tag{1}$$

where $\ell^t$ is the loss function on the accumulated all class set, $K = |\bigcup_{i=1}^{t-1} \mathcal{Y}^i|$ is the number of previous classes, the $\lambda$ is a balance hyper-parameter, and $\mathcal{B}_j$ is the buffer for the $j$-th class. The second term ensures the representative training nodes from previous classes are included in the current training phase, efficiently mitigating the risk of catastrophic forgetting of previous classes. Also, the size of $\mathcal{B}_j$ is much smaller than the total node number of class $j$. Thus, the number of training nodes required is significantly lower than joint training, leading to a substantial increase in efficiency.

## 3 METHODOLOGY

Without loss of generality, we choose the plain GCN model followed by a classifier head as the backbone of $f(\theta)$, which encodes each node $v_i$ in graphs into an embedding $z_i$ and a probability $p_i$. As shown in Figure 2, initially, this model was trained on the graph $G_1$. When the $t$-th task introducing new classes arrives, we first extend new parameters (highlighted as the yellow segment) into the last layer of the classifier, ensuring the output probabilities encompass the previous and newly introduced classes. To facilitate continual training of $f(\theta)$, we leverage the memory replay

framework, which incorporates both the Heuristic Diversified Memory Selection (Section 3.1) and the Diversified Memory Generation Replay (Section 3.2).

**Motivation**. Consider $p(G_{<t})$ as the true data distribution of graphs from prior $t-1$ tasks, and $q(\mathcal{B}_{<t})$ as the data distribution encapsulated within the memory replay buffers $\mathcal{B}$ sampled from $G_{<t}$. To understand the efficacy of the buffer diversity in incremental learning scenarios, we engage in a theoretical examination to indicate that a high diversity within $\mathcal{B}$ ensures that the empirical loss $\mathcal{L}(\theta)$ over $\mathcal{B}$ closely mirrors the total expected loss $\mathcal{L}(\theta)$ over $p(G_{<t})$.

**Theorem 1.** *Let the loss function $\mathcal{L}(\theta, x)$ be $\beta$-Lipschitz continuous in respect to the input $x$. Under this condition, the discrepancy between the expected loss under the true data distribution $p(G_{<t})$ and that under the replay buffer distribution $q(\mathcal{B}_{<t})$ is bounded as follows:*

$$\left| \mathbb{E}_{v \sim p(G_{<t})}[\mathcal{L}(\theta, v)] - \mathbb{E}_{v \sim q(\mathcal{B}_{<t})}[\mathcal{L}(\theta, v)] \right| \leq \beta \cdot W(p(G_{<t}), q(\mathcal{B}_{<t})), \tag{2}$$

*where $W(p, q)$ denotes the Wasserstein distance between distributions $p$ and $q$, defined by: $W(p, q) = \inf_{\gamma \in \Gamma(p,q)} \mathbb{E}_{(v,v') \sim \gamma}[d(v, v')]$, and $\Gamma(p, q)$ represents the set of all possible joint distributions (couplings) that can be formed between $p$ and $q$.*

Assume that both $p(G_{<t})$ and $q(\mathcal{B}_{<t})$ follow Gaussian distributions with means $\mu_p, \mu_q$ and covariance matrices $\Sigma_p, \Sigma_q$ respectively. Thus, the squared 2-Wasserstein distance between two Gaussian distributions is given by:

$$W_2^2(\mathcal{N}(\mu_p, \Sigma_p), \mathcal{N}(\mu_q, \Sigma_q)) = \|\mu_p - \mu_q\|^2 + \text{Tr}(\Sigma_p + \Sigma_q - 2(\Sigma_p^{1/2} \Sigma_q \Sigma_p^{1/2})^{1/2}). \tag{3}$$

Since the $\Sigma$ measure the distribution diversity and $\mathcal{B}_{<t}$ is the subset of $\mathcal{G}_{<t}$ and is typically less diverse (detailedly analyzed in Appendix A.1). Assuming the sampling strategy is unbiased upon means, as the $\mathcal{B}_{<t}$ more diversified, $\Sigma_q \to \Sigma_p$, leading to the Wasserstein distance decreases. Based on Theorem 1, the discrepancy between the expected loss under true distribution and the buffer distribution becomes less, making the optimization on the buffer more closely approximate the optimization on all previous graph data.

## 3.1 HEURISTIC DIVERSIFIED MEMORY SELECTION

Based on the above motivations. For memory selection, to ensure that the selected nodes are adequately diverse with respect to the classification task, we consider the two perspectives: **P1**: *the nodes within the same buffer should exhibit sufficient diversity to faithfully represent disparate regions of their corresponding areas*. Also, **P2**: *the inter-class distance between nodes residing in distinct buffers should be maximized to facilitate the model to delineate clear classification boundaries*. Thus, we introduce the concepts of intra-diversity and inter-diversity for the buffers. Our goal is to select the buffer $\mathcal{B}_i$ corresponding to the $i$-th class of training nodes based on the following criteria:

$$\mathcal{B}_i = \arg\max_{\mathcal{B}_i \subset \mathcal{C}_i} \sum_{v \in \mathcal{B}_i} [\underbrace{\mathcal{A}(v, \mathcal{B}_i)}_{\text{intra-diversity}} + \underbrace{\frac{1}{K-1} \sum_{j=1, j \neq i, \mathcal{B}_j \subset \mathcal{C}_j}^{K} \mathcal{A}(v, \mathcal{B}_j)}_{\text{inter-diversity}}], \tag{4}$$

where $\mathcal{C}_i$ is set of $i$-th class of training nodes, $\mathcal{A}(v, \mathcal{B}_i, G^t)$ denotes the distance measure between node $v$ and $\mathcal{B}_i$ in the current graph $G^t$, which we define as the L2-norm distance on probabilities between node $v$ and its closest node in $\mathcal{B}_i$. While the measure can be defined as any topological distance, such as the shortest path, we use probability distance because it offers finer resolution, reduced noise towards tasks, and computational efficiency. The first term quantifies the intra-diversity within the buffer $\mathcal{B}_i$, reflecting the variations among its own nodes, while the second term quantifies the inter-diversity between $\mathcal{B}_i$ and other buffers, illustrating the differences between the nodes of $\mathcal{B}_i$ and those belonging to other buffers.

**Heuristic Greedy Solution.** However, achieving this objective for selecting different classes of buffers is an NP-hard problem. This kind of problem is usually addressed using heuristic methods (Hochbaum, 1996). Thus, we introduce a greedy algorithm to sample representative training nodes when new tasks are introduced. Specifically, suppose $\mathcal{V}^t = \bigcup_{i=K+1}^{K+n} \mathcal{C}_i^t$ is the training nodes of $t$-th task and $\mathcal{C}_i^t$ is the training set corresponding to the $i$-th novel class. We have previously selected buffers $\{\mathcal{B}_i\}_{i=1}^{K}$ in previous $t-1$ tasks, where $K$ and $n$ are the numbers of previous classes and

novel classes, respectively. Then, the greedy selection strategy is defined in the Algorithm 1, where $\mathbb{D}(\mathcal{B}_i) = \sum_{v \in \mathcal{B}_i} (\mathcal{A}(v, \mathcal{B}_i) + \frac{1}{K-1} \sum_{j=1, j \neq i, \mathcal{B}_j \subset \mathcal{C}_j}^{K} \mathcal{A}(v, \mathcal{B}_j))$ is the set score function defined on the buffer set $\mathcal{B}_i$ of the $i$-th class, $\triangle_{\mathbb{D}}(v|\mathcal{B}_i)$ is the gain of $f$ choosing $v$ into $\mathcal{B}_i$, and $v*$ is the chosen node using the greedy strategy. In greedy Algorithm 1, the core idea is to make the currently best choice of buffer nodes at every step, hoping to obtain the global optimal solution for the objective Eq.4 through this local optimal choice.

---

**Algorithm 1: Heuristic Buffer Selection**

---

**Input:** $\{\mathcal{B}_i\}_{i=1}^{K}$. // buffers of previous tasks.
   $\{\mathcal{C}_i^t\}_{i=K+1}^{K+n}$. // training node sets of novel classes of current tasks.
   $f(\theta)$. // Model after trained on the $(t-1)$-th task.
**Output:** $\{\mathcal{B}_i\}_{i=1}^{K+n}$ // updated buffers.
   /* Initializing. */
1 Create empty buffer $\{\mathcal{B}_i\}_{i=K+1}^{K+n}$.
2 **for** $i$ *from* $K+1$ *to* $K+n$ **do**
3     Select one node with highest output probability on the $i$-th label from $\mathcal{C}_i^t$ into $\mathcal{B}_i$ via $f(\theta)$.
4 **end**
   /* Greedy Selecting. */
5 **repeat**
6     **for** $i$ *from* $K+1$ *to* $K+n$ **do**
7        $\triangle_{\mathbb{D}}(v|\mathcal{B}_i) = \mathbb{D}(\mathcal{B}_i \cup v) - \mathbb{D}(\mathcal{B}_i)$,
8        $v^* = \arg\max_{\mathcal{C}_i \setminus \mathcal{B}_i} \triangle_{\mathbb{D}}(v|\mathcal{B}_i)$.
9        Add $v^*$ into $\mathcal{B}_i$.
10     **end**
11 **until** $b$ *nodes are sampled in each buffer*;

---

Below, we give a Proposition of approximation guarantee of our greedy algorithm.

**Proposition 1.** *(Greedy Approximation Guarantee of Algorithm 1). The greedy Algorithm 1 that sequentially adds elements to an initially empty set based on the largest marginal gain $\triangle_{\mathbb{D}}$ under a cardinality constraint provides a solution $\mathcal{B}_i^*$ that is at least $(1 - \frac{1}{e})$ times the optimal solution, i.e.,*

$$f(\mathcal{B}_i^*) \geq \left(1 - \frac{1}{e}\right) \cdot f(OPT), \qquad (5)$$

*where $OPT$ represents the optimal solution of the buffer set $\mathcal{B}_i$.*

*Proof.* The above Proposition can be derived from the Greedy Approximation Guarantee for Monotonic and Submodular Functions (Nemhauser et al., 1978) (proof can be found in Theorem 2 in the Appendix), given that our function $f$ is both monotonic (from Lemma 1) and submodular (from Lemma 2). The greedy algorithm is guaranteed to produce a solution that is at least $(1 - \frac{1}{e})$ times the optimal solution. $\quad\square$

**Time Complexity Analysis.** For each buffer of the $t$-th task, there are $b$ sampling steps where $b$ is the size of the buffers. Each sampling can be done in $O((K+n) * |\mathcal{C}_i^t|)$, by determining distances and making comparisons. Thus, the overall complexity of selecting each buffer is $O(b(K+n) * |\mathcal{C}_i^t|)$. Note that $b$ (the buffer size) and $K + n$ (the total number of classes up to $t$-th task) are typically much smaller than $|\mathcal{C}_i^t|$, ensuring the efficiency of the algorithm.

## 3.2 DIVERSIFIED MEMORY GENERATIVE REPLAY

During training for the $t$-th task $\mathcal{T}^t$, the stored representative nodes in the buffers $\{\mathcal{B}_i\}_{i=1}^{K}$ from previous $t-1$ tasks are also recalled to reinforce what the model has previously learned, known as memory replay. However, the limited buffer size still presents challenges: **C1**: *the stored knowledge may be constrained and might not encompass the full complexity of previous tasks, leading to a potential bias in replaying*, and **C2**: *the training process can become difficult as the model may easily overfit to the limited nodes in the buffers, undermining its ability to generalize across different tasks.* Thus, we proposed the Diversified Memory Generation Replay to address the above problems.

**Broadening Diversity of Buffer Node Embeddings**. Specifically, the embeddings of the buffer nodes are first subjected to a variational layer, which aims to create more nuanced representations that encapsulate the inherent probabilistic characteristics of nodes. Let $z_i \in \mathbb{R}^h$ denote the embedding of node $v_i \in \mathcal{B}_j, 1 \leq j \leq K$, where $h$ is the hidden dimension. Specifically, we treat the nodes in the buffers as the observed samples $\mathcal{V}_{ob}$ drawn from the ground-truth distribution of the previous nodes:

$$Z = \bigcup_{j=1}^{K} \bigcup_{v_i \in \mathcal{B}_j} \{z_i\} \in \mathbb{R}^{(K \times b) \times h} \stackrel{\text{def}}{=\joinrel=} \mathcal{V}_{ob}. \qquad (6)$$

The node variable $\widehat{Z}$ is drawn from the variational network layer $q_{\phi_v}(\widehat{Z}|Z)$ with parameters $\phi_v$. Specifically, $q_{\phi_v}$ outputs the mean and variance of the node embeddings distributions respectively,

expressed as $Z^\mu = q^\mu_{\phi_v}(Z)$ and $Z^\sigma = q^\sigma_{\phi_v}(Z)$, where $q^\mu_{\phi_v}(\cdot)$ is the identity function and $q^\sigma_{\phi_v}(\cdot)$ is a Liner layer followed with a Relu activation layer. Then we use the reparameterization technique to sample from $\mathcal{N}(Z^\mu, Z^\sigma)$, expressed as $\widehat{Z} = Z^\mu + Z^\sigma \odot \epsilon$, where $\widehat{Z} \in \mathbb{R}^h$ and $\epsilon$ is drawn from standard normal distributions. Thus, we define the generated samples as $\mathcal{V}_{ge}$. The variational operation augments the diversity of observed buffer nodes $\mathcal{V}_{ob}$, empowering the model to explore more expansive distribution spaces.

**Maintaining Integrity of Synthesized Embeddings.** Then, we further propose to maintain the integrity of these variational node embeddings to prevent them from deviating too far, which implies that while the generated embeddings should exhibit diversity, they must remain similar to the ground-truth ones. Specifically, we adopt an adversarial learning strategy. We introduce an auxiliary discriminator, $\mathcal{D} : \mathbb{R}^h \to \mathbb{R}^1$ with parameters $\phi_d$, tasked with distinguishing the original embeddings and the variational embeddings of nodes. In contrast, the model is learned to generate diversified variational node embeddings from the original ones, meanwhile ensuring the authenticity of synthesized variational node embeddings. Thus, the learning objective is defined as follows:

$$\mathcal{L}_{MISE} = \min_{\theta, \phi_v} \mathbb{E}_{z_i \sim Z} \mathbb{E}_{q_{\phi_v}(\hat{z}_i | z)} \left[ \max_{\phi_d} \ \ell_\mathcal{D}(z_i, \hat{z}_i) \right], \tag{7}$$

where $\ell_\mathcal{D}$ is a negative binary cross-entropy loss function on the variational and original node embeddings, defined as:

$$\ell_\mathcal{D}(z_i, \hat{z}_i) = \log \mathcal{D}(z_i, \phi_d) + \log\left(1 - \mathcal{D}(\hat{z}_i, \phi_d)\right). \tag{8}$$

The min-max adversarial learning strategy involves training the domain discriminator to distinguish whether the node embeddings are synthesized or original while simultaneously enforcing a constraint on the model to generate indistinguishable node embeddings from the domain discriminator. This interplay aims to yield synthesized node embeddings that are more comprehensive and maintain integrity. It leverages the strengths of generative methods for increased representational complexity of buffer nodes to address **C1**. Simultaneously, it employs adversarial learning and regularization to ensure this expansion does not lead to distortions.

**Consolidating Generalization of Synthesized Embeddings.** Furthermore, to adequately capture the node relationships within the variational node embeddings, we use the variational node embeddings to generate a reconstructed graph on the buffer nodes. As the buffer nodes are sampled from disparate regions, and the initial connections between them are sparse, we instead employ the ground-truth label to build the reconstructed graph. Specifically, nodes sharing the same labels are linked, while those with different labels are not connected. The reconstructed graph is denoted as $\widehat{A} \in \mathbb{R}^{Kb \times Kb}$ and the decoder loss is defined as:

$$\begin{aligned} \mathcal{L}_{CGSE} &= -\mathbb{E}_{q_{\phi_v}(\widehat{Z}|Z)}[\log p(\widehat{A}|\widehat{Z})] + \mathrm{KL}(q_{\phi_v}(\hat{z}_i|z) || p(\hat{z}_i)) \\ &= -\mathbb{E}_{\hat{z}_i, \hat{z}_j \sim \widehat{Z}} \left[ \widehat{A}_{ij} \log \widehat{p}_{ij} + \widehat{A}_{ij} \log(1 - \widehat{p}_{ij}) \right] + \mathrm{KL}(q_{\phi_v}(\hat{z}_i|z) || p(\hat{z}_i)), \end{aligned} \tag{9}$$

where $p(\widehat{A}|\widehat{Z})$ is the probability of reconstructing $\widehat{A}$ given the latent variational node embedding matrix $\widehat{Z}$, following a Bernoulli distribution. $\widehat{p}_{ij}$ is the probability of an edge between nodes $i$ and $j$, defined as: $\widehat{p}_{ij} = \mathrm{sigmoid}(\hat{z}_i^T \cdot \hat{z}_j)$. The second term in Eq. 9 is a distribution regularization term, which enforces the variational distribution $q_{\phi_v}(\hat{z}_i|z)$ of each node to be close to a prior distribution $p(\hat{z}_i)$, which we assume is a standard Gaussian distribution. $\mathrm{KL}(\cdot)$ represents the Kullback-Leibler (KL) divergence. The detailed derivation of $\mathcal{L}_{CGSE}$ is in the Appendix.

The reconstruction objective incorporates both the inter-class and intra-class relationships between nodes in buffers. This loss facilitates the learning of variational node embeddings with well-defined classification boundaries, further bolstering the model's generalization on the label space. As such, the method can effectively address **C2**.

**Replaying on Generated Diversified Memory.** Finally, we define the reply objective on the variational embeddings of buffer nodes, rather than the original embeddings, expressed as:

$$\mathcal{L}_{RP} = \sum_{j=1}^{K} \sum_{v_i \in \mathcal{B}_j} \ell^t(\hat{z}_i, y_i). \tag{10}$$

Note that the variational operation regenerates synthetic buffer node embeddings with the same size as original embeddings in each training step, i.e., $|\mathcal{V}_{ge}| \equiv |\mathcal{V}_{ob}|$, broadening the diversity of memory while guaranteeing the efficiency of the buffer replay.

## 3.3 OVERALL OPTIMIZATION

Combining the new task loss $\mathcal{L}^t = \sum_{v_i \in \mathcal{V}^t} \ell^t(f(v_i; \theta), y_i)$ and the above memory replay losses, the overall optimization objective can be written as follows:

$$\min_{\theta} \mathcal{L}^t + \min_{\theta, \phi_v} \left\{ \lambda_1 \mathcal{L}_{RP} + \lambda_2 \max_{\phi_d} \{\mathcal{L}_{MISE}\} + \lambda_3 \mathcal{L}_{CGSE} \right\}, \tag{11}$$

where $\lambda_1$, $\lambda_2$ and $\lambda_3$ is the loss weights.

**Synchronized Min-Max Optimization.** A gradient reversal layer (GRL) (Ganin et al., 2016) is introduced between the variational embedding and the auxiliary discriminator so as to conveniently perform min-max optimization on $\theta, \phi_v$ and $\phi_d$ under $\mathcal{L}_{MISE}$ in the same training step. GRL acts as an identity transformation during the forward propagation and changes the signs of the gradient from the subsequent networks during the backpropagation.

**Scaliability on Large-Scale graphs.** In each training step, training DMSG on the entire graph and buffer at once may not be practical, especially for large-scale graphs. Following methods like GraphSAGE (Hamilton et al., 2017) and GraphSAINT (Zeng et al., 2019), we adopt a mini-batch optimization strategy. We sample a multi-hop neighborhood for each node and set two kinds of batch sizes, $B^{\text{new}}$ and $B^{\text{buffer}}$, for the new task and replay losses in Eq. 11, respectively. The ratio of these batch sizes corresponds to the ratio between the total training nodes in the new task and the buffer.

## 4 EXPERIMENTS

**Experiment Setup.** In this section, we describe the experiments we perform to validate our proposed method. We use the four growing graph datasets, CoraFull, OGB-Arxiv, Reddit, and OGB-Products, introduced in Continual Graph Learning Benchmark (CGLB) (Zhang et al., 2022a). These graphs contain 35, 20, 20, and 23 sub-graphs, respectively, where each sub-graph corresponds to new tasks with novel classes. For baselines, we establish the upper bound baseline **Joint** defined in Section 2. The lower bound baseline **Fine-tune** employs only the newly arrived training nodes for model adaptation without memory replay. Then, we set multiple continual learning models for the graph as baselines, including EWC (Kirkpatrick et al., 2017), MAS (Aljundi et al., 2018), GEM (Lopez-Paz & Ranzato, 2017), LwF (Li & Hoiem, 2017), TWP (Liu et al., 2021), ER-GNN (Zhou & Cao, 2021), SSM (Zhang et al., 2022c), and SEM (Zhang et al., 2023). For a fair comparison, the backbone is set as two layers and the hidden dimension as 256 for all baselines. For evaluation metric, we first report the accuracy matrix $M^{acc} \in \mathbb{R}^{T \times T}$, which is lower triangular where $M_{i,j}^{acc}(i \geq j)$ represents the accuracy on the $j$-th tasks after learning the task $i$. To derive a single numeric value after learning all tasks, we report the Average Accuracy (AA) $\frac{1}{T} \sum_{i=1}^{T} M_{T,i}^{acc}$ and the Average Forgetting (AF) $\frac{1}{T-1} \sum_{i=1}^{T-1} M_{T,i}^{acc} - M_{i,i}^{acc}$ for each task after learning the last task. For a more detailed introduction to the experimental setup, please refer to Appendix A.5.

**Overall Comparison.** This experiment aims to answer: *How is DMSG's performance on the continual learning on graphs?* We compare DMSG with various baselines in the class-incremental continual learning task and report the experimental results in Table 1. Initially, we observe that DMSG attains a significant margin over other baseline methods across all datasets. Certain baseline methods demonstrate exceedingly poor results. This can be attributed to the difficulty of the problem, which involves more than 20 timesteps' continual learning. When the model forgets intermediate tasks, errors are cumulatively compounded for subsequent tasks, potentially leading to the model easily collapsing. However, our model addresses this challenge through superior buffer selection and replay training strategies, effectively avoiding catastrophic problems. Among the various baselines, the most comparable method to DMSG is SEM. Our method outperforms SEM mainly because we improve the buffer selection strategy, i.e., instead of random selection, we employ distance measures to choose more representative nodes for each class. Additionally, the variational replay method enables our model to effectively learn the data distribution from previous tasks. When compared to Joint, DMSG achieves comparable results, demonstrating its effectiveness in preserving knowledge from previous tasks, even with limited training samples. Notably, our method outperforms Joint on the Reddit dataset, and also exhibits a positive AF. This improvement can be attributed to the proposed buffer selection strategy, which can select representative nodes, in other words, eliminate the noise nodes, and thereby enhance the results.

Table 1: The model performance comparisons( ↑: higher is better, Joint is the upper bound).

| Methods | CoreFull | | OGB-Arxiv | | Reddit | | OGB-Products | |
|---|---|---|---|---|---|---|---|---|
| | AA/% ↑ | AF/% ↑ | AA/% ↑ | AF/% ↑ | AA/% ↑ | AF/% ↑ | AA/% ↑ | AF/% ↑ |
| Fine-tune | 3.5±0.5 | -95.2±0.5 | 4.9±0.0 | -89.7±0.4 | 5.9±1.2 | -97.9±3.3 | 3.4±0.8 | -82.5±0.8 |
| EWC | 52.6±8.2 | -38.5±12.1 | 8.5±1.0 | -69.5±8.0 | 10.3±11.6 | -33.2±26.1 | 23.8±3.8 | -21.7±7.5 |
| MAS | 12.3±3.8 | -83.7±4.1 | 4.9±0.0 | -86.8±0.6 | 13.1±2.6 | -35.2±3.5 | 16.7±4.8 | -57.0±31.9 |
| GEM | 8.4±1.1 | -88.4±1.4 | 4.9±0.0 | -89.8±0.3 | 28.4±3.5 | -71.9±4.2 | 5.5±0.7 | -84.3±0.9 |
| TWP | 62.6±2.2 | -30.6±4.3 | 6.7±1.5 | -50.6±13.2 | 13.5±2.6 | -89.7±2.7 | 14.1±4.0 | -11.4±2.0 |
| LwF | 33.4±1.6 | -59.6±2.2 | 9.9±12.1 | -43.6±11.9 | 86.6±1.1 | -9.2±1.1 | 48.2±1.6 | -18.6±1.6 |
| ER-GNN | 34.5±4.4 | -61.6±4.3 | 30.3±1.5 | -54.0±1.3 | 88.5±2.3 | -10.8±2.4 | 56.7±0.3 | -33.3±0.5 |
| SSM | 75.4±0.1 | -9.7±0.0 | 48.3±0.5 | -10.7±0.3 | 94.4±0.0 | -1.3±0.0 | 63.3±0.1 | -9.6±0.3 |
| SEM | 77.7±0.8 | -10.0±1.2 | 49.9±0.6 | -8.4±1.3 | 96.3±0.1 | -0.6±0.1 | 65.1±1.0 | -9.5±0.8 |
| Joint | 81.2±0.4 | -3.3±0.8 | 51.3±0.5 | -6.7±0.5 | 97.1±0.1 | -0.7±0.1 | 71.5±0.1 | -5.8±0.3 |
| **DMSG** | **77.8±0.3** | **-0.5±0.5** | **50.7±0.4** | **-1.9±1.0** | **98.1±0.0** | **0.9±0.1** | **66.0±0.4** | **-0.9±1.6** |

Figure 3: Dynamics of the average accuracy during incremental learning on different growing graphs.

**In-Depth Analysis of Continuous Performance.** This experiment aims to answer: *How does DMSG's fine-grained performance evolve after continuously learning each task?* To present a more fine-grained demonstration of the model's performance in continual learning on graphs, we analyzed the average performance across all previous tasks each time a new task was learned. The comparative results of Fine-tune, Joint, DMSG, and the top-performing baseline, SEM, are depicted in Figure 3. The curve represents the model's performance after $t$ in terms of AA on all previous $t$ tasks. Also, we visualize the accuracy matrices of DMSG and SEM on the OGB-Arxiv and OGB-Product datasets. The results are presented in Figure 4. In these matrices, each row represents the performance across all tasks upon learning a new one, while each column captures the evolving performance of a specific task as all tasks are learned sequentially. In the visual representation, lighter shades signify better performance, while darker hues indicate inferior outcomes. From the results, we observed that as the number of tasks increases, the learning objectives grow increasingly complex, resulting in a reduction in performance across all examined methods, including Joint. That is because as tasks accumulate and the learning objectives become multifaceted, it becomes challenging for models to maintain optimal performance across all classes. Notably, the Fine-Tuning strategy experienced a substantial decline, with the model collapsing with the arrival of merely two new tasks, demonstrating that catastrophic forgetting occurs almost immediately when the model fails to access previous memories. This reinforces the need for effective continual learning techniques on the growing graphs where new tasks frequently emerge. While the performance drop was observed across all methods, DMSG demonstrated resilience and outperformed the top-performing baseline SEM. Also, DMSG predominantly displays lighter shades across the majority of blocks compared to SEM in Figure 4. Moreover, its competitive performance with Joint in specific datasets signifies its robustness and capability. This could be attributed to diversified memory selection and generation in DMSG that not only help in mitigating forgetting but also in adapting efficiently to new tasks.

**Component Analysis of Memory Replay.** This experiment aims to answer: *Are all the proposed memory replay technologies of DMSG have the claimed contribution to continual learning?* To investigate the distinct contributions of the diversified memory generation replay method, we conducted an ablation study on it. We design three variant methods for DMSG— **w/o** $\mathcal{L}_{MISE}$: This variant excludes the adversarial learning loss for maintaining the integrity of synthesized embeddings; **w/o** $\mathcal{L}_{CGSE}$: This variant excludes the graph reconstruction loss on variational embeddings for consolidating their generalization to label space; **w/o** all: both losses were removed, and as a result, the model operates without any variational embeddings and just replays the original nodes in the memory buffers. From the results in Table 2, we can observe when both $\mathcal{L}_{MISE}$ and $\mathcal{L}_{CGSE}$ are removed (w/o all), the performance is the lowest across all datasets. Also, a progressive improvement is observed as individual components in the model. This confirms their respective contributions to continual learning. An interesting trend emerges when comparing the individual contributions of the two components.

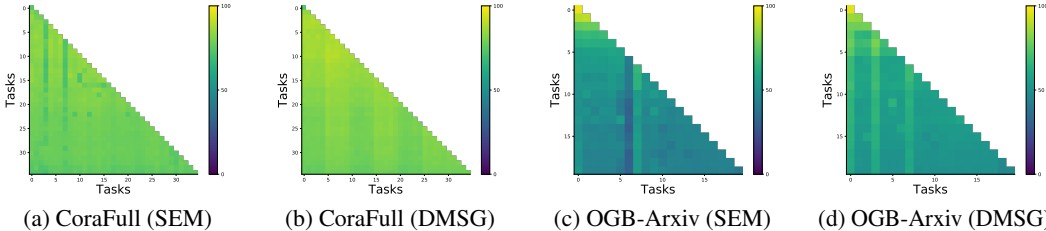

| (a) CoraFull (SEM) | (b) CoraFull (DMSG) | (c) OGB-Arxiv (SEM) | (d) OGB-Arxiv (DMSG) |

Figure 4: Accuracy matrices of DMSG and SEM in different datasets.

The w/o $\mathcal{L}_{CGSE}$ variant slightly surpasses the performance of w/o $\mathcal{L}_{MISE}$. This suggests that while both components are crucial, adversarial variational embedding learning may have a more pronounced effect in capturing essential and diverse patterns inherent in the data. The best performance occurs with all components, supporting that the proposed components are beneficial individually and collectively, ensuring the model can effectively memorize the previous knowledge while continually adapting to new tasks.

Table 2: The ablation study of memory replay.

| Methods | CoreFull | OGB-Arxiv | Reddit | OGB-Products |
|---|---|---|---|---|
| w/o all | 73.9±0.6 | 48.2±0.4 | 89.3±3.6 | 60.1±0.8 |
| w/o $\mathcal{L}_{MISE}$ | 74.4±0.8 | 49.3±0.3 | 95.1±2.9 | 60.1±0.9 |
| w/o $\mathcal{L}_{CGSE}$ | 74.8±0.7 | 49.7±0.2 | 97.8±0.4 | 60.5±0.7 |
| DMSG | **77.8±0.3** | **50.7±0.4** | **98.1±0.0** | **66.0±0.4** |

## 5 RELATED WORKS

Many graphs in real-world applications, such as social networks and transportation systems, are not static but evolve over time. To accommodate this dynamic nature, various methods have been developed to manage growing graph data (Wang et al., 2020a; Tang & Matteson, 2020; Daruna et al., 2021; Luo et al., 2020). **Incremental learning** (Parisi et al., 2019; Schwarz et al., 2018; Castro et al., 2018) involves models continuously learning and adapting, often facing the catastrophic forgetting problem. Solutions include regularization techniques (Pomponi et al., 2020; Lin et al., 2023), parameter isolation (Wang et al., 2021a; Lyu et al., 2021), and memory replay (Wang et al., 2021b; Mai et al., 2021). Recent works (Lu et al., 2022; Wang et al., 2022; Yang et al., 2022; Tan et al., 2022) specifies this problem to growing graph data–new nodes introduce unseen classes to the graph. Methods of **Graph Incremental Learning** (Rakaraddi et al., 2022; Kim et al., 2022; Sun et al., 2023; Su & Wu, 2023; Feng et al., 2023; Liu et al., 2023; Niu et al., 2024) strive to retain knowledge of current classes and adapt to new ones, enabling continuous prediction across all classes. For example, TWP (Liu et al., 2021) employs regularization to ensure the preservation of critical parameters and intricate topological configurations, achieving continuous learning. HPNs (Zhang et al., 2022b) adaptively choose different trainable prototypes for incremental tasks. ER-GNN (Zhou & Cao, 2021) proposes multiple memory sampling strategies designed for the replay of experience nodes. SEM (Zhang et al., 2023) leverages a sparsified subgraph memory selection strategy for memory replay on growing graphs. However, the trade-off between buffer size and replay effect is still a Gordian knot, i.e., aiming for a small buffer size usually results in *ineffective memory preserving and knowledge replay*. To address this gap, this paper introduces an effective memory selection and replay method that explores and preserves the essential and diversified knowledge contained within restricted nodes, thus improving the model in learning previous knowledge.

## 6 CONCLUSION

To summarize, this paper presents a novel approach DMSG to the challenge of incremental learning in ever-growing and increasingly complex graph-structured data. Central to memory diversification, the proposed method includes a holistic and efficient buffer selection module and a generative memory replay module to effectively prevent the model from forgetting previous tasks when learning new tasks. The proposed method works in both preserving comprehensive knowledge in limited memory buffers and enhancing previous knowledge memorization when learning new tasks.

One potential limitation of DMSG is that it does not improve the graph feature extractor of the model, which may result in suboptimal performance when dealing with increasing graph data, as the model parameters are insufficient to learn and retain massive amounts of information effectively. Future work may focus on integrating more sophisticated parameter incremental learning techniques to dynamically adapt the model to the growing complexity of graph data, ultimately leading to improved performance in incremental learning scenarios.

## 7 ACKNOWLEDGMENT

The work is partially supported by the National Natural Science Foundation of China (Grant No. 62406056), the Beijing Natural Science Foundation (Grant No. 4254089), the National Key R&D Program of China (Grant No. 2023YFF0725001), the National Natural Science Foundation of China (Grant No. 92370204), the Guangdong Basic and Applied Basic Research Foundation (Grant No. 2023B1515120057), and Guangzhou-HKUST(GZ) Joint Funding Program (Grant No. 2023A03J0008), Education Bureau of Guangzhou Municipality. The computational resources are supported by SongShan Lake HPC Center (SSL-HPC) in Great Bay University.

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

# A APPENDIX

## A.1 EXTENDED ANALYSIS OF THEOREM 1

In the context of Theorem 1 and Eq.3. The covariance matrices $\Sigma_p$ and $\Sigma_q$ are key indicators of the diversity within the distributions of $p(G_{<t})$ and $q(B_{<t})$, respectively. According to the Cochran theorem (Soch, 2023), the sampled variance $\sigma_q^2$ for an univariate is related to the chi-squared distribution:

$$(n-1)\frac{\sigma_q^2}{\sigma_p^2} \sim \chi_{n-1}^2$$

where $\sigma_p$ is the ture variance and $n$ is the sample number. The mean of the chi-squared distribution is $n-1$, and its right-skewed nature implies that:

$$P(\chi_{n-1}^2 < n - 1) > 0.5$$

$$P(\sigma_q^2 < \sigma_p^2) = P\left((n-1)\frac{\sigma_q^2}{\sigma_p^2} < n - 1\right) = P(\chi_{n-1}^2 < n - 1) > 0.5$$

This indicates that there is a greater than 50% chance that the sample variance underestimates the population variance. In higher dimensions, the sample covariance matrix $\Sigma_q$ (related to the Wishart distribution) displays properties analogous to the chi-squared distribution used in the unidimensional case. This statistical property extends to the sample covariance matrix in higher dimensions, suggesting that the eigenvalues of $\Sigma_q$ are likely to be smaller than those of $\Sigma_p$.

Thus, the replay buffer $B_{<t}$, being a sampled subset of $\mathcal{G}_{<t}$, typically exhibits less diversity than the entire graph dataset from previous tasks, meaning $\Sigma_q$ is often smaller than $\Sigma_p$. Assuming the sampling strategy is unbiased upon means, as the $\mathcal{B}_{<t}$ more diversified, $\Sigma_q \to \Sigma_p$, leading to the Wasserstein distance decreases. Based on Theorem 1, the discrepancy between the expected loss under true distribution and the buffer distribution becomes less, making the optimization on the buffer more closely approximate the optimization on all previous graph data.

## A.2 THEORETICAL ANALYSIS OF GREEDY ALGORITHM 1.

In the Heuristic Diversified Memory Selection section, we propose a heuristic buffer selection algorithm and give a proposition of approximation guarantee of the algorithm:

**Proposition 2.** *(Greedy Approximation Guarantee of Algorithm 1). The greedy Algorithm 1 that sequentially adds elements to an initially empty set based on the largest marginal gain $\triangle_{\mathbb{D}}$ under a cardinality constraint provides a solution $\mathcal{B}_i^*$ that is at least $(1 - \frac{1}{e})$ times the optimal solution, i.e.,*

$$f(\mathcal{B}_i^*) \geq \left(1 - \frac{1}{e}\right) \cdot f(OPT), \tag{12}$$

*where $OPT$ represents the optimal solution of the buffer set $\mathcal{B}_i$.*

Below we first give a greedy approximation guarantee and then give the proof of Proposition 1.

**Theorem 2.** *(Greedy Approximation Guarantee for Monotonic and Submodular Functions): Let $f : 2^U \to \mathbb{R}$ be a set function defined on a finite ground set $U$ such that:*

- $f$ is non-decreasing (monotonic), i.e., for every $A \subseteq B \subseteq U$,

$$f(A) \leq f(B)$$

- $f$ is submodular, i.e., for every $A \subseteq B \subseteq U$ and for every $x \in U \setminus B$, the marginal increase in $f$ due to $x$ is at least as large when added to the smaller set $A$ as when added to the larger set $B$. Formally,

$$f(A \cup \{x\}) - f(A) \geq f(B \cup \{x\}) - f(B)$$

*Then, a greedy algorithm that sequentially adds elements to an initially empty set based on the largest marginal gain of $f$ produces a solution $S^*$ such that:*

$$f(S^*) \geq \left(1 - \frac{1}{e}\right) \cdot f(OPT)$$

*where $OPT$ is an optimal solution. Below is the proof:*

*Proof.* Let $S$ be the set constructed by the greedy algorithm. Begin with $S = \emptyset$ and $f(S) = 0$.

At the $k$-th step, the greedy algorithm picks an element $x_k$ that maximizes the marginal gain:

$$x_k = \arg \max_{x \in U \setminus S} \Delta_f(x|S)$$

where $\Delta_f(x|S) = f(S \cup \{x\}) - f(S)$.

Let $OPT$ be an optimal solution, and without loss of generality, let $OPT = \{o_1, o_2, \ldots, o_m\}$. For each $k$, consider the gain of the greedy algorithm in the $k$-th step relative to adding the $k$-th element of $OPT$ to $S$:

$$\Delta_f(x_k|S) \geq \frac{1}{m} \sum_{i=1}^{m} \Delta_f(o_i|S)$$

This inequality is derived from the submodularity of $f$. The right-hand side is the average marginal gain of adding elements of $OPT$ to $S$.

The total increase in $f$ over the first $k$ steps of the greedy algorithm is at least:

$$f(S) \geq \left(1 - \left(1 - \frac{1}{m}\right)^k\right) \cdot f(OPT)$$

By analyzing the expression on the right and considering the limit as $k$ approaches $m$, we obtain the $\left(1 - \frac{1}{e}\right)$ factor.

After $m$ steps (or fewer, if the greedy algorithm terminates early), the set $S^*$ constructed by the greedy algorithm satisfies:

$$f(S^*) \geq \left(1 - \frac{1}{e}\right) \cdot f(OPT)$$

This completes the proof. $\square$

Given $\mathbb{D}(\mathcal{B}_i) = \sum_{v \in \mathcal{B}_i}(\mathcal{A}(v, \mathcal{B}_i) + \frac{1}{K-1} \sum_{j=1, j \neq i}^{K} \mathcal{A}(v, \mathcal{B}_j))$ is the set score function defined on the buffer set $\mathcal{B}_i$ of the $i$-th class, $\triangle_{\mathbb{D}}(v|\mathcal{B}_i)$ is the gain of $f$ choosing $v$ into $\mathcal{B}_i$. We have the following Lemmas of $\mathbb{D}$:

**Lemma 1.** *The function $\mathbb{D}$ is monotonic with respect to set $\mathcal{B}_i$.*

*Proof.* For the intra-class diversity, when new nodes are added to $\mathcal{B}_i$, they positively contribute to $\mathbb{D}$ due to their own distances to other nodes within $\mathcal{B}_i$. The addition of these nodes can decrease the distances of the existing nodes in $\mathcal{B}_i$ to their closest neighbors. For the inter-class diversity, the addition of new nodes to $\mathcal{B}_i$ increases the value of $\mathbb{D}$ since these nodes have their own distances to nodes in other classes $\mathcal{B}_j$. The distances from existing nodes in $\mathcal{B}_i$ to nodes in other classes remain unchanged, so there is no loss in inter-class diversity. Given that the gains from inter-class distances generally overshadow the potential losses from intra-class distances, we can infer that as we include more nodes in $\mathcal{B}_i$, the value of $\mathbb{D}$ will increase. Therefore, the function is monotonic. $\square$

**Lemma 2.** *The function $\mathbb{D}$ is submodular with respect to set $\mathcal{B}_i$.*

*Proof.* For the function $\mathbb{D}$, to be submodular, it must satisfy the following condition: For any set $\mathcal{B}_i$ and node $v$, and any set $\mathcal{V}^-$ from $\mathcal{C}_i$ such that $v$ is not an element of $\mathcal{V}^-$, the following inequality holds:

$$\mathbb{D}(\mathcal{B}_i \cup v) - \mathbb{D}(\mathcal{B}_i) \geq \mathbb{D}(\mathcal{B}_i \cup \mathcal{V}^- \cup v) - \mathbb{D}(\mathcal{B}_i \cup \mathcal{V}^-)$$

For the first term $\sum_{v \in \mathcal{B}_i} \mathcal{A}(v, \mathcal{B}_i)$, given the definition of the function $\mathcal{A}$, the distance to the nearest point in $\mathcal{B}_i$ for any $v$ will be greater than or equal to the distance to the nearest point in $\mathcal{B}_i \bigcup \mathcal{V}^-$. Therefore, adding $v$ to $\mathcal{B}_i$ can potentially reduce the distance more significantly than adding it to $\mathcal{B}_i \bigcup \mathcal{V}^-$. Regarding the subsequent term, which sums the distances from all nodes in the set to other sets of buffers $\mathcal{B}_j$, it can be deduced that adding $v$ into either $\mathcal{B}_i$ or $\mathcal{B}_i \bigcup \mathcal{V}^-$ yields an identical gain in the function's value. In light of the above observations, we can validate the aforementioned inequality. Consequently, the function $\mathbb{D}$ is submodular with respect to set $\mathcal{B}_i$. $\square$

Eventually, our optimal solution in Equation 4 can be written as:

$$\mathcal{B}_i = \arg \max_{\mathcal{B}_i \subset \mathcal{C}_i} \mathbb{D}(\mathcal{B}_i)$$

The proposed greedy Algorithm 1 aims to sequentially add elements to $\mathcal{B}_i$ based on the largest marginal gain of $\mathbb{D}$, which is monotonic and submodular. Based on Theorem 2, Lemma 1, and Lemma 2, we can establish the validity of Proposition 1.

### A.3 DERIVATION OF LOSS $\mathcal{L}_{CGSE}$.

In the Diversified Memory Generation Replay section, we aim to use the variational node embeddings to generate a reconstructed graph on the buffer nodes. The optimization problem is described by the likelihood of observing a reconstructed adjacency matrix $\widehat{A}$ based on surrogate source node embeddings $\hat{Z}$:

**Theorem 3.** *Given the variational posterior $q(\hat{z}_i|z)$ via the instantiating of synthesized node embeddings and a prior distribution $p(\hat{z}_i)$ acting as a regularization for $q(\hat{z}_i|z)$, we have the Evidence Lower Bound (ELBO), which is a surrogate objective for maximizing the log-likelihood:*

$$\log p(\widehat{A}) \geq \mathbb{E}_{q_{\phi_v}(\widehat{Z}|Z)}[\log p(\widehat{A}|\widehat{Z})] - \mathrm{KL}(q_{\phi_v}(\hat{z}_i|z)||p(\hat{z}_i))$$

where $\mathrm{KL}(\cdot)$ represents the Kullback-Leibler (KL) divergence. The term $p(\widehat{A}|\widehat{Z})$ denotes the reconstruction probability from the surrogate source node embeddings to the reconstructed adjacency matrix.

*Proof.* The optimization problem we are dealing with is described by the likelihood of observing a reconstructed adjacency matrix $\widehat{A}$ based on surrogate source node embeddings $\widehat{Z}$:

$$\log p(\widehat{A}) = \log \int p(\widehat{A}, \widehat{Z}) d\widehat{Z}$$
$$= \log \int p(\widehat{A}|\widehat{Z}) p(\widehat{Z}) d\widehat{Z}$$

This equation encapsulates the joint probability of observing the graph and the latent variables. To enable tractable optimization, we introduce a variational distribution, $q_{\phi_v}(\hat{z}_i|z)$, approximating the true posterior of the node embeddings. Multiplying and dividing by this term, we can express the likelihood as:

$$\log p(\widehat{A}) = \log \int p(\widehat{A}|\widehat{Z}) \frac{p(\widehat{Z})}{q_{\phi_v}(\hat{z}_i|z)} q_{\phi_v}(\hat{z}_i|z) d\widehat{Z}$$

This expression can be interpreted as an expectation with respect to the variational distribution:

$$\log p(\widehat{A}) = \log \mathbb{E}_{q_{\phi_v}}\left[p(\widehat{A}|\widehat{Z})\frac{p(\widehat{Z})}{q_{\phi_v}(\hat{z}_i|z)}\right]$$

Applying Jensen's inequality allows us to bring the logarithm inside the expectation, leading to a lower bound:

$$\log p(\widehat{A}) \geq \mathbb{E}_{q_{\phi_v}}\left[\log p(\widehat{A}|\widehat{Z})\frac{p(\widehat{Z})}{q_{\phi_v}(\hat{z}_i|z)}\right]$$

$$= \mathbb{E}_{q_{\phi_v}}\left[\log p(\widehat{A}|\widehat{Z})\right] + \mathbb{E}_{q_{\phi_v}}\left[\log \frac{p(\widehat{Z})}{q_{\phi_v}(\hat{z}_i|z)}\right]$$

We recognize the second term in the above inequality as the negative of the KL divergence between the variational distribution and the prior:

$$\text{KL}(q_{\phi_v}(\hat{z}_i|z)||p(\hat{z}_i)) = \mathbb{E}_{q_{\phi_v}}[\log \frac{q_{\phi_v}(\hat{z}_i|z)}{p(\hat{z}_i)}]$$

The KL divergence acts as a regularization term, penalizing deviations of our variational distribution from the prior. Combining the terms, we have the Evidence Lower Bound (ELBO), which is a surrogate objective for maximizing the log-likelihood:

$$\log p(\widehat{A}) \geq \mathbb{E}_{q_{\phi_v}}[\log p(\widehat{A}|\widehat{Z})] - \text{KL}(q_{\phi_v}(\hat{z}_i|z)||p(\hat{z}_i))$$

Thus, we obtain the optimization objective $\mathcal{L}_{CGSE}$. Specifically, the actual calculation formulas of the two components of $\mathcal{L}_{CGSE}$ are defined as:

$$\mathbb{E}_{q_{\phi_v}}[\log p(\widehat{A}|\widehat{Z})] = \left[\widehat{A}_{ij}\log \widehat{p}_{ij} + \widehat{A}_{ij}\log(1 - \widehat{p}_{ij})\right]$$

$$\text{KL}(q_{\phi_v}(\hat{z}_i|z)||p(\hat{z}_i)) = \sum_{k=1}^{K}\text{KL}\left(\mathcal{N}(\mu_k, \Sigma_k)||\mathcal{N}(0, I)\right)$$

$$= \sum_{k=1}^{K}\frac{1}{2}\left(\text{tr}(\Sigma_k) + \mu_k^{\mathsf{T}}\mu_k - h - \log|\Sigma_k|\right)$$

where $tr(\cdot)$ denotes the trace of a matrix, $h$ is the dimensionality of the source node embedding $\hat{z}_i$, and $|\Sigma_k|$ is the determinant of $\Sigma_k$. □

## A.4 SYNCHRONIZED MIN-MAX OPTIMIZATING OF OVERALL LOSS $\mathcal{L}_{DMSG}$.

Recall that the overall optimization objective is as follows:

$$\min_{\theta}\mathcal{L}^t + \min_{\theta,\phi_v}\left\{\lambda_1\mathcal{L}_{RP} + \lambda_2\max_{\phi_d}\{\mathcal{L}_{MISE}\} + \lambda_3\mathcal{L}_{CGSE}\right\},$$

where $\lambda_1$, $\lambda_2$ and $\lambda_3$ is the weights to balance different losses. Then, the parameters $\theta$, $\phi_v$, and $\phi_d$ of DMSG can be optimized by Stochastic Gradient Decent (SGD) as follows:

$$\theta \leftarrow \theta - \mu\left(\frac{\partial\mathcal{L}^t}{\partial\theta} + \lambda_1\frac{\partial\mathcal{L}_{RP}}{\partial\theta} + \lambda_2\frac{\partial\mathcal{L}_{MISE}}{\partial\theta} + \lambda_3\frac{\partial\mathcal{L}_{CGSE}}{\partial\theta}\right),$$

$$\phi_v \leftarrow \phi_v - \mu\left(\lambda_1\frac{\partial\mathcal{L}_{RP}}{\partial\phi_v} + \lambda_2\frac{\partial\mathcal{L}_{MISE}}{\partial\phi_v} + \lambda_3\frac{\partial\mathcal{L}_{CGSE}}{\partial\phi_v}\right),$$

$$\phi_d \leftarrow \theta - \mu\left(-\lambda_2\frac{\partial\mathcal{L}_{MISE}}{\partial\phi_d}\right).$$

where $\mu$ is the learning rate.

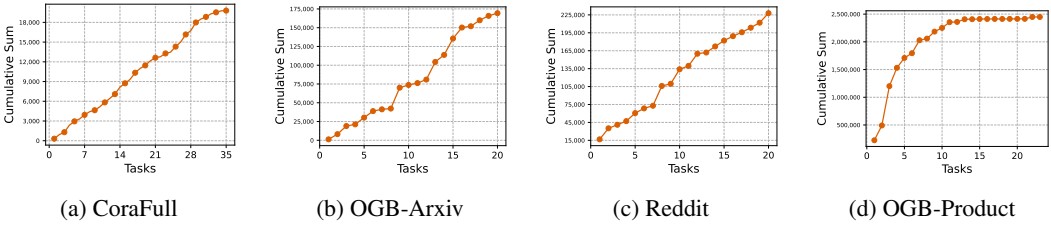

| (a) CoraFull | (b) OGB-Arxiv | (c) Reddit | (d) OGB-Product |

Figure 5: Cumulative number of nodes within different growing graphs.

Table 3: The statistics of four datasets.

| Dataset | #Nodes | #Edges | #Classes | #Tasks |
|---|---|---|---|---|
| CoreFull | 19,793 | 130,622 | 70 | 35 |
| OGB-Arxiv | 169,343 | 1,166,243 | 40 | 20 |
| Reddit | 232,965 | 114,615,892 | 40 | 20 |
| OGB-Products | 2,449,029 | 61,859,140 | 46 | 23 |

## A.5 EXPERIMENT SETUP

### A.5.1 DATASETS

We use the four graph datasets, CoraFull, OGB-Arxiv, Reddit, and OGB-Products, introduced in Continual Graph Learning Benchmark (CGLB) (Zhang et al., 2022a). CoraFull (McCallum et al., 2000) and OGB-Arxiv (Wang et al., 2020b) are citation networks with papers as nodes and citation relationships as edges and labeled based on paper topics. Reddit (Hamilton et al., 2017) is a social network with posts as nodes and posts are connected if the same user comments. The node labels are the community the posts belong to. OGB-Products (Hu et al., 2020) is a product co-purchasing network with nodes representing products and edges indicating that the connected products are purchased together. The node labels are the class the products belong to. Each graph is considered a volume-increasing graph over time, where its size expands with the arrival of new nodes with several novel classes at each time, leading to a more complex node classification task. To elaborate, given an original graph comprised of $C$ class nodes, these classes are partitioned into $m = \frac{C}{k}$ groups with the original class order. This ensures that each group contains $k$ classes of nodes. The graph is divided into $m$ sub-graphs accordingly, where each sub-graph represents the newly added data with novel classes at a particular time, resulting in a new learning task. In the context of continual learning, we commence by training a model on the first graph. Subsequently, at each time, we integrate the succeeding sub-graph into the prevailing graph, prompting the model to continuously learn and adapt to new tasks. To accentuate the forgetting problems in continual learning, we set $k$ as 2 for each graph, which basically maximizes the number of tasks on a growing graph, challenging the capabilities of baselines. The detailed statistics of the datasets are shown in Table 3. Figure 5 shows the curves of the cumulative number of nodes within different growing graphs..

### A.5.2 BASELINES

First, we establish the upper bound and lower bound baselines of our problem. The upper bound baseline **Joint** is defined in Section 2, which involves continuously training the model each time with all accumulated training nodes from previous and new tasks, thus without the forgetting problem. The lower bound baseline **Fine-tune** employs only the newly arrived training nodes for model adaptation, yielding a fine-tuning mode that easily forgets the previous tasks. Then, we set multiple continual learning models for graph as baselines, including EWC (Kirkpatrick et al., 2017), MAS (Aljundi et al., 2018), GEM (Lopez-Paz & Ranzato, 2017), LwF (Li & Hoiem, 2017), TWP (Liu et al., 2021), ER-GNN (Zhou & Cao, 2021), SSM (Zhang et al., 2022c), and SEM (Zhang et al., 2023). A detailed introduction to these methods can be found in the Appendix. As some of the methods are not originally designed for graph data, we utilize a Graph Neural Network (GNN) as the backbone of all baselines for extracting graph information. The specific GNN model of baselines is selected from

GCN (Welling & Kipf, 2016), SGC (Wu et al., 2019), GAT (Casanova et al., 2018), and GIN (Xu et al., 2018), based on which yields the best performance. For a fair comparison, the backbone is set as two layers and the hidden dimension as 256 for all baselines. Here, we give a detailed introduction to the baseline models.

- **EWC** imposes a constraint, based on the Fisher Information Matrix and weight adjustments, to preserve knowledge from prior tasks during new task learning.

- **LwF** learns new tasks without necessitating the retention of old task samples by training neural networks to minimize discrepancies between predictions on new and preceding tasks.

- **GEM** facilitates effective continual learning by storing knowledge from previous tasks and archiving prior tasks while permitting updates solely if they don't amplify the loss on these tasks.

- **MAS** mitigates catastrophic forgetting by imposing penalties on modifications to critical synapses, determined by the influence of weight changes on output, and preserves old knowledge without requiring the storage of previous tasks.

- **TWP** preserves both the minimized loss and the topological structure of the graph, and harmonizes past and new task knowledge, thus enabling more robust model performance and circumventing catastrophic forgetting.

- **SSM** uses a strategy that sparsifies computational graphs into a fixed size before storing them in memory, which not only minimizes memory consumption but also enhances the learning of tasks from diverse classes.

- **SEM** develops the subgraph episodic memory to store the explicit topological information in the form of computation subgraphs and perform memory replay-based continual graph representation learning.

### A.5.3 EXPERIMENTAL SETTING

We use a 2-layer GCN as our backbone. For the baselines, we report their experimental results from SEM (Zhang et al., 2022c), and we set the same experimental setting with the bucket size of DMSG as 60 for CoraFull and 400 for other datasets. For the diversified memory replay, we set the number of generated diversified node embeddings in the memory buffers the same as the original counts. Following (Zhang et al., 2022a;c), the training rate for each task is set as 60%, the validation rate as 20%, and the test rate as 20%. The training epoch for each task is set as 200 epochs. The training nodes of previous tasks exclusively come from the buffer. The learning rate is set as 0.001 and the weight decay as 1e-3, The weights for the four losses are set as $[1, 20, 1, 1]$, respectively. For evaluation metric, we utilize the Average Accuracy (AA) meaning the mean accuracy of the model on all tasks after learning the final task, and Average Forgetting (AF) meaning the average reduction in accuracy for each task from when it was first learned to after the model has learned all tasks. we first report the accuracy matrix $M^{acc} \in \mathbb{R}^{T \times T}$, which is lower triangular where $M_{i,j}^{acc}(i \geq j)$ represents the accuracy on the $j$-th tasks after learning the task $i$. To derive a single numeric value after learning all tasks, we report the Average Accuracy (AA) $\frac{1}{T} \sum_{i=1}^{T} M_{T,i}^{acc}$ and the Average Forgetting (AF) $\frac{1}{T-1} \sum_{i=1}^{T-1} M_{T,i}^{acc} - M_{i,i}^{acc}$ for each task after learning the last task.

### A.6 EXTENTAL EXPERIMENTS

### A.6.1 VISUALIZATION OF ACCURACY MATRIX

To further understand the dynamics of our methods under the incremental learning task, we visualize the accuracy matrices of DMSG and SEM on the CoraFull and OGB-Arxiv datasets. The results are presented in Figure 4 and 6. In these matrices, each row represents the performance across all tasks upon learning a new one, while each column captures the evolving performance of a specific task as all tasks are learned sequentially. In the visual representation, lighter shades signify better performance, while darker hues indicate inferior outcomes. Upon comparison, DMSG predominantly displays lighter shades across the majority of blocks compared to SEM, showing that DMSG can consistently achieve better performance.

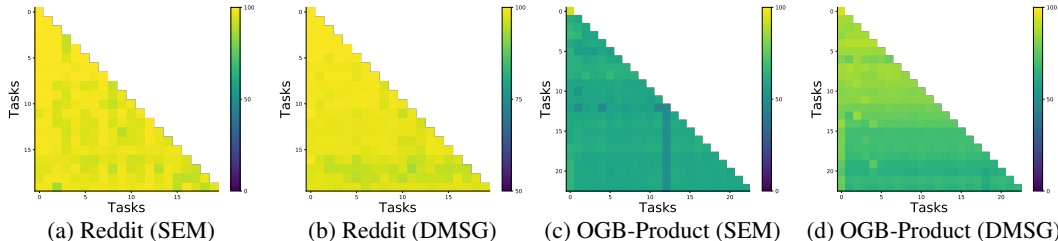

| (a) Reddit (SEM) | (b) Reddit (DMSG) | (c) OGB-Product (SEM) | (d) OGB-Product (DMSG) |

Figure 6: Accuracy matrices of DMSG and SEM in different datasets..

### A.6.2 ANALYSIS OF BUFFER SELECTION.

This experiment aims to answer: *Is the proposed buffer selection strategy more effective than other selection strategies in selecting representative nodes?* The choice of buffer sampling strategies is critical in a graph's continual learning, as it determines which knowledge will be stored and later replayed to the model. We choose different buffer sampling strategies, including our proposed Diversified Memory Selection (DMS) method, K-center sampling (Nguyen et al., 2018), Coverage Maximization (CM), and Mean of Feature (MF) (Zhou & Cao, 2021). We perform these sampling methods in our continual learning framework. The results are demonstrated in Table 4. We can observe that our sampling method consistently excels in performance. This consistency can be attributed to exploring both the intra-class and inter-class diversity of nodes within our method that efficiently captures the required diversity among nodes, ensuring a holistic representation of the dataset. The K-center approach, although it shows commendable results, lags behind our method. This can be attributed to it sharing a similar assumption with our method–capturing diverse sample buffers. However, the K-center approach does not fully consider intra- and inter-class diversity, , leading to comparatively lower performance. It's worth highlighting that both CM and MF, when implemented within our framework, demonstrate a marked improvement in performance compared to their native ER-GNN framework. This outcome underscores the flexibility of our proposed diversified memory generation replay model, suggesting that it not only complements various sampling strategies but can potentially improve their effectiveness.

Table 4: Comparison of buffer selection methods.

| **Methods** | CoreFull | OGB-Arxiv | Reddit | OGB-Products |
|---|---|---|---|---|
| Kcenter | 74.4±0.7 | 43.7±0.6 | 85.2±2.7 | 49.7±0.8 |
| CM | 74.5±0.6 | 40.6±1.0 | 72.4±1.4 | 43.5±0.8 |
| MF | 74.5±0.8 | 42.3±0.9 | 87.2±0.7 | 49.3±1.1 |
| **DMS** | **77.8±0.3** | **50.7±0.4** | **98.1±0.0** | **66.0±0.4** |

### A.6.3 ANALYSIS OF BUFFER SIZES.

This experiment aims to answer: *How do different buffer size affect the performance of DMSG?* To comprehensively understand the influence of buffer size on the performance of DMSG, we evaluate DMSG across different buffer sizes, specifically 40, 50, 60, 70, and 80 for the CoraFull dataset, and 200, 300, 400, 500, and 600 for other datasets. The results are illustrated in Figure 7. There's a clear trend that, generally, as the buffer size increases, the performance also sees an enhancement. This can be intuitively understood: a more extensive buffer can store more data from past tasks, acting as a richer source of knowledge when learning new tasks and thereby mitigating the effects of catastrophic forgetting. An equally important observation is the consistency in the standard deviation of results across different buffer sizes. The compact standard deviation, even for smaller buffers, is indicative of the robustness of DMSG. On both the Reddit and OGB-Product datasets, the performance at a buffer size of 600 exhibited a marginal decline. While it might appear counterintuitive given the general trend, this could be due to the larger buffer size introducing more noise nodes. These nodes, which might not be as representative or essential as others, can dilute the quality of information stored. As a result, they consequently reduce the model's generalization ability-—a consideration that aligns with our discussion in the overall comparison.

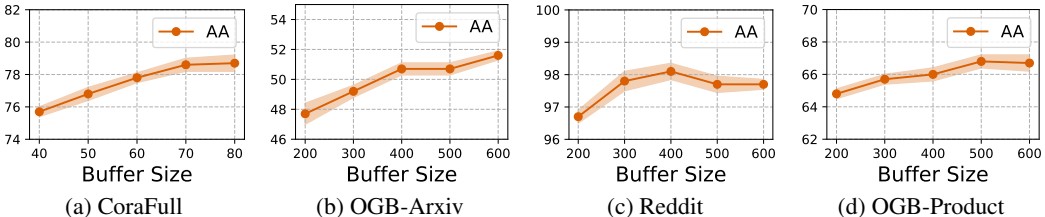

Figure 7: The model performance with different buffer sizes.

## A.7 DETAILED RELATED WORKS

### A.7.1 LEARNING ON GROWING GRAPHS.

Graph-based learning often operates under the assumption that the entire graph structure is available upfront. For example, Graph Neural Networks (GNNs) (Ju et al., 2024; Qiao et al., 2020) have rapidly become one of the most prominent tools for learning graph-structured data, bridging the gap between deep learning and graph theory. Representative methods includes GCN (Welling & Kipf, 2016), GraphSAGE (Hamilton et al., 2017), and GAT (Veličković et al., 2018), etc. However, these methods predominantly operate on static graphs. Many graphs in real-world applications, such as social networks and transportation systems, are not static but evolve over time. To accommodate this dynamic nature, various methods have been developed to manage growing graph data (Wang et al., 2020a; Tang & Matteson, 2020; Daruna et al., 2021; Luo et al., 2020). For instance, Evolving Graph Convolutional Networks (EvolveGCN) (Pareja et al., 2020) emphasizes temporal adaptability in graph evolution. Temporal Graph Networks (TGNs) (Rossi et al., 2020) operates on continuous-time dynamic graphs represented as a sequence of events. Spatio-Temporal Graph Networks (STGN)(Yu et al., 2018) integrates spatial and temporal information to enhance prediction accuracy. However, these methods primarily concentrate on a singular task in evolving graphs and often encounter difficulties when more complicated tasks emerge as the graph expands.

### A.7.2 INCREMENTAL LEARNING.

Incremental learning (Parisi et al., 2019) refers to a evolving paradigm within machine learning where the model continues to learn and adapt after initial training. The continuous integration of new tasks often leads to the catastrophic forgetting problem. There are usually two types of continual learning settings–class-incremental learning is about expanding the class space within the same task domain, while task-incremental learning involves handling entirely new tasks, which may or may not be related to previous ones (Schwarz et al., 2018; Castro et al., 2018). Typically, three different categories of methods have emerged to address the continual learning problem. The first category revolves around regularization techniques (Pomponi et al., 2020; Lin et al., 2023). By imposing constraints, these methods prevent significant modifications to model parameters that are critical to previous tasks, ensuring a degree of stability and retention. The second category encompasses parameter-isolation-based approaches (Wang et al., 2021a; Lyu et al., 2021). These strategies dynamically allocate new parameters exclusively for upcoming tasks, ensuring that crucial parameters intrinsic to previous tasks remain unscathed. Lastly, memory replay-based methods (Wang et al., 2021b; Mai et al., 2021) present a solution by selectively replaying representative data from previous tasks to mitigate the extent of catastrophic forgetting, while are more preferred due to their reduced memory storage requirements and flexibility in parameter training. This paper delve deeper and present an effective strategy for selecting and replaying memory on graphs.

### A.7.3 GRAPH CLASS-INCREMENTAL LEARNING.

Graph incremental learning, also known as graph continual learning, aims to continually train a graph model on growing graphs to perform more complex tasks. This problem includes two settings, task-incremental learning and class-incremental learning. Graph task incremental learning methods treat the learning as different tasks and assume the task affectations of newly added nodes are known when inference. Many works (Su & Wu, 2023; Su et al., 2023; Niu et al., 2023; Su et al., 2024) have been introduced recently under this setting. Class-incremental Learning (Belouadah et al., 2021) means

the specific scenario where the number of classes increases along with the new samples introduced, which does not presuppose knowledge of node class assignments in different tasks, presenting a more complex challenge. Graph class-incremental learning (Lu et al., 2022; Wang et al., 2022; Yang et al., 2022; Tan et al., 2022) specifies this problem to growing graph data–new nodes introduce unseen classes to the graph. Methods of graph class-incremental learning (Rakaraddi et al., 2022; Kim et al., 2022; Sun et al., 2023; Feng et al., 2023; Liu et al., 2023; Niu et al., 2024) strive to retain knowledge of current classes and adapt to new ones, enabling continuous prediction across all classes. In the past years, various strategies have been proposed to tackle this intricate problem. For example, TWP (Liu et al., 2021) employs regularization to ensure the preservation of critical parameters and intricate topological configurations, achieving continuous learning. HPNs (Zhang et al., 2022b) adaptively choose different trainable prototypes for incremental tasks. ER-GNN (Zhou & Cao, 2021) proposes multiple memory sampling strategies designed for the replay of experience nodes. SSM (Zhang et al., 2022c) and SEM (Zhang et al., 2022c) leverage a sparsified subgraph memory selection strategy for memory replay on growing graphs. PDGNN (Niu et al., 2023) proposes parameter decoupled graph neural networks with topology-aware embedding memory for the graph incremental learning problem. PUMA (Liu et al., 2024) and DeLoMe (Zhang et al., 2024) both use different graph condensation technology to preserve memory graphs. However, the trade-off between buffer size and replay effect is still a Gordian knot, i.e., aiming for a small buffer size usually results in ineffective knowledge replay. To address this gap, this paper introduces an effective memory selection and replay method that explores and preserves the essential and diversified knowledge contained within restricted nodes, thus improving the model in learning previous knowledge.

