# OpenReview forum: "Towards Continuous Reuse of Graph Models via Holistic Memory Diversification"
_ICLR.cc/2025/Conference — ICLR 2025 Poster_

### Official Review · Reviewer_T2ad · 2024-10-27

**Soundness:** 2
**Presentation:** 3
**Contribution:** 2
**Rating:** 6
**Confidence:** 5

**Summary:**

This paper presents a memory replay approach to address the challenges of graph incremental learning. Specifically, the proposed method includes a diversified buffer selection module and a generative memory replay module to prevent the model from forgetting previous tasks when learning new tasks. The experimental results demonstrate the effectiveness of the proposed method.

**Strengths:**

The studied problem is practical and interesting.

The paper is well-written and easy to understand.

The proposed method introduces novel approaches for buffer selection and memory utilization.

**Weaknesses:**

1. The title presented on the website is not the same as the one in the paper
2. The citation style is not used appropriately. Most of the citations should be in parenthesis using \citep{}.
3. The authors are encouraged to include the time consumption of different memory selection strategies.
4. The statements in lines 178-181 are not clear. Please revise it to make it clearer.
5. The definition of L_{adv} is not clear in line 334. Moreover, there is no Eq.A.3.
6. The analysis of parameter sensitivity is missing.
7. The comparisons to more recent baselines are encouraged.

**Questions:**

Please see the weaknesses.

---

> ### Author Response · Authors · 2024-11-24
>
> > **W1: The title presented on the website is not the same as the one in the paper**
>
> We are sorry for the clerical error. We initially considered multiple titles and inadvertently failed to replace the title in the latest version of the manuscript. We will ensure this is corrected.
>
> > **W2: The citation style is not used appropriately. Most of the citations should be in parenthesis using \citep{}.**
>
> Thank you for pointing out the inconsistencies in our citation style. We will revise all citations to ensure they are in parentheses using \citep{} as required by the citation guidelines.
>
>
> > **W3: The authors are encouraged to include the time consumption of different memory selection strategies.**
>
> In our paper, we compared our proposed heuristic diversified memory selection methods with three other memory selection strategies--K-center sampling, Coverage Maximization (CM), and Mean of Feature (MF). Here we analyze the time complexity to assess the efficiency of different methods.
>
> For our method, for each batch, there are $b$ sampling steps where $b$ is the size of the buffers. Each sampling can be done in $O((K+n)|\mathcal{C}^t_{i}|)$, by determining distances and making comparisons, where $K+n$ is the total number of classes up to $t$-th task and $\mathcal{C}^t_i$ is the training set corresponding to the $i$-th class.
> Thus, the overall complexity of selecting each buffer is $O(b(K+n)*|\mathcal{C}^t_{i}|)$.
>
> For K-center sampling, the time complexity of selecting $b$ clustering centers for each class is $O(b*|\mathcal{C}^t_{i}|)$. Thus, the overall complexity is also $O(bK*|\mathcal{C}^t_{i}|)$.
>
> For Coverage Maximization and Mean of Feature, they did not report the time complexity in their own paper, we further analyze them based on their papers and codes. Coverage Maximization needs to calculate and sort the distance of nodes in each class to other classes, which costs at least $O(K(K-1)|\mathcal{C}^t_{i}|^2)$ where $|\mathcal{C}^t_{i}|$ is the mean of node number in each class, which could much more complex.  Mean of Feature calculates the prototypes of each class and selects nodes are closest to the prototype, the time complexity is $O(K|\mathcal{C}^t_{i}| + K|\mathcal{C}^t_{i}|log(|\mathcal{C}^t_{i}|))$.
>
> Considering the node number in each class is much more than the buffer size and class number, it is evident that our heuristic diversified memory selection method, while being slightly more complex compared to K-center sampling, remains more efficient than the rest. From the comparison results in **Table 4 in Section A.5.2**, our method achieves much better results, offering a reasonable trade-off between computational efficiency and effective diversity enhancement of the memory buffer.
>
>
> > **W4: The statements in lines 178-181 are not clear. Please revise it to make it clearer.**
>
> Sorry if we did not make it clear.
>
> Here's a more comprehensive version that better explains the statements in lines 178-181:
>
> The covariance matrices $\Sigma_p$ and $\Sigma_q$ are key indicators of the diversity within the distributions of $p(G_{<t})$ and $q(B_{<t})$, respectively. Typically, the replay buffer $B_{<t}$, being a sampled subset of $G_{<t}$, exhibits less diversity than the entire graph dataset from previous tasks, meaning $\Sigma_q$ is often smaller than $\Sigma_p$.
> Assuming that the sampling strategy used to populate $B_{<t}$ is unbiased with respect to the mean of the distributions, enhancing the diversity of $B_{<t}$ so that it better represents the full dataset, $\Sigma_q$ will approach $\Sigma_p$. This leads to a smaller Wasserstein distance because the distributions are more similarly "shaped.":
>
> $W_2^2(\mathcal{N}(\mu_p, \Sigma_p), \mathcal{N}(\mu_q, \Sigma_q)) = \| \mu_p - \mu_q \|^2 + \operatorname{Tr}(\Sigma_p + \Sigma_q - 2(\Sigma_p^{1/2} \Sigma_q \Sigma_p^{1/2})^{1/2}).$
>
> According to the bound in Theorem 1:
>
> $\left| E_{v \sim p(G_{<t})}[\mathcal{L}(\theta, v)] -E_{v \sim q(B_{<t})}[\mathcal{L}(\theta, v)] \right| \leq \beta \cdot W(p(G_{<t}), q(B_{<t}))$
>
> A decrease in the Wasserstein distance leads to a smaller discrepancy between the expected loss under the true distribution $ p(G_{<t}) $ and the distribution represented by the replay buffer $ q(B_{<t}) $. Consequently, optimizing the model using the replay buffer becomes a closer approximation to optimizing using the entire dataset from previous tasks, thereby improving the effectiveness of incremental learning.
>
> We will revise the statements to be more clear in the manuscript.

---

> > ### Author Response · Authors · 2024-11-24
> >
> > > **W5: The definition of L_{adv} is not clear in line 334. Moreover, there is no Eq.A.3.**
> >
> > Thank you for pointing out. We will correct the loss term to $L_{MISE}$ and update the equation reference to Eq. 11 in the manuscript.
> >
> >
> > > **W6: The analysis of parameter sensitivity is missing.**
> >
> > We have conducted hyperparameter sensitivity experiments on the critical buffer size in **Section A.5.3** to explore how the number of nodes in the buffer affects our model's performance. The results of these experiments are presented in **Figure 7** of our manuscript. Our findings indicate that increasing the buffer size tends to improve the model’s ability to retain knowledge, particularly for more complex datasets, but up to a point where the returns diminish. This variation study helps in understanding the trade-offs involved in buffer size selection concerning memory constraints and performance optimization.
> >
> >
> > > **W7: The comparisons to more recent baselines are encouraged.**
> >
> > We appreciate the suggestion! We are sorry for the short time to conduct new experiments to include more baselines. We have included 8 baselines, providing a comprehensive analysis within the current scope. Thank you for your understanding and we will certainly consider including more comparisons in subsequent studies based on your valuable feedback.

---

> > > ### Comment · Reviewer_T2ad · 2024-11-24
> > >
> > > I appreciate the clarification, which has addressed most of my concerns. I will maintain my positive score.

---

> > > > ### Author Response · Authors · 2024-11-25
> > > >
> > > > We are glad our clarification was helpful and appreciate your positive feedback. Thank you for your constructive comments to strengthen our paper.

---

### Official Review · Reviewer_Mh5c · 2024-11-03

**Soundness:** 2
**Presentation:** 3
**Contribution:** 2
**Rating:** 6
**Confidence:** 3

**Summary:**

This paper studies incremental learning on growing graphs, aiming to continually train a model on new tasks without loosing the inference ability on previous tasks. The method proposed in this work, namely holistic diversified memory selection and generation (DMSG), is a memory based technique, with a focus on improving the data diversity in the memory, which is neglected by the existing works. The unique design of DMSG comes from two perspectives. First, it proposes a novel method to sample the data, second, instead of directly using the buffered data, DMSG train a generator to generate the data to replay.

Overall, this method proposes a novel memory based technique, and the experimental results justify the effectiveness of the method. However, there are also some major problems regarding the theoretical analysis, as listed below in the weakness part.

**Strengths:**

1. The paper is overall well written, and the main idea is clearly conveyed.

2. Experiments are comprehensively conducted on multiple benchmark datasets against multiple baselines.

3. According to the results, the proposed method consistently outperform the baselines, although the improvement on some datasets are negligible.

**Weaknesses:**

1. The theoretical analysis is not rigorous enough. Theorem 1 reveals that smaller distance between the memory data distribution and the full data distribution indicates that the loss computed based on memory can more closely mirrors the expected over the full previous data. This is a straightforward and reasonable conclusion. But what follows is not convincing enough. First, it is assumed that the distributions of the memory data and full data is Gaussian. I guess this is acceptable, although a little bit arbitrary. Next, it is stated that the buffered data is typically less diverse, therefore increasing its diversity could make it closer to the full data distribution. This is not convincing, since the 'diversity' here refers to the covariance. Why is the buffered data has a smaller variance than the real (full) distribution? This should depends on the sampling strategy instead of the size of the set. Moreover, although the diversity here refers to the covariance, in the following of the paper, the diversity is measured by the average distance among the data, which is inconsistent.

2. The implemented diversity maximization strategy is inconsistent with the theoretical motivation, as mentioned in the first weakness point.

**Questions:**

1. Please explain more on the probability distance used in Section 3.1, e.g. the specific formulation.

2. Is the replay data generator trained from scratch each time when a new task comes in? Or is the generator train each time when a new task comes in, but is initialized from the generator from the previous task?

3. If the generator is trained every time when a new task comes in, will this induce significant extra computational burden?

---

> ### Author Response · Authors · 2024-11-22
>
> > **W1: The theoretical analysis is not rigorous enough. Theorem 1 reveals that smaller distance between the memory data distribution and the full data distribution indicates that the loss computed based on memory can more closely mirrors the expected over the full previous data. This is a straightforward and reasonable conclusion...**
>
> Thank you for your constructive feedback.
>
> The assumption that the data follow Gaussian distributions is a common simplification in theoretical analyses. This assumption is justified by the Central Limit Theorem[1] under certain conditions.
>
> In the context of Theorem 1, 'diversity' refers specifically to the spread of data as captured by covariance matrices. This is a measure of how broadly the data are distributed around the mean, which is directly relevant to how well the buffer represents the variability seen in the full data.
> The mention that buffered data is "typically less diverse" is indeed subject to sampling variability, particularly when the sample size is small. Here is a simple prove:
>
> In the univariate case, the sample variance $\sigma_q^2$ is related to the chi-squared distribution[2] as follows:
>
> $(n-1)\frac{\sigma_q^2}{{\sigma_p}^2}\sim \chi^2_{n-1}$
>
> where $\sigma_p$ is the ture variance and $n$ is the sample number. The mean of the chi-squared distribution is $n-1$, and its right-skewed nature implies that:
>
> $P(\chi^2_{n-1} < n-1) > 0.5$, i.e.,
>
> $P(\sigma_q^2 < \sigma_p^2) = P\left( (n-1)\frac{\sigma_q^2}{\sigma_p^2} < n-1 \right) = P(\chi^2_{n-1} < n-1) > 0.5$
>
> This indicates that there is a greater than 50% chance that the sample variance underestimates the population variance.  In higher dimensions, the sample covariance matrix (related to the Wishart distribution) display properties analogous to the chi-squared distribution used in the unidimensional case. This statistical property extends to the sample covariance matrix in higher dimensions, suggesting that the eigenvalues of $\Sigma_q$ are likely to be smaller than those of $\Sigma_p$. This supports our initial claim--"typically less diverse" from a high-dimensional perspective.
>
>
>
>
> [1] https://en.wikipedia.org/wiki/Central_limit_theorem
>
> [2] https://statproofbook.github.io/P/norm-chi2.html
>
>
>
> > **W2: The implemented diversity maximization strategy is inconsistent with the theoretical motivation, as mentioned in the first weakness point.**
>
> Thank you for pointing out this.
>
> The theoretical foundation of our approach was intended to emphasize the importance of a diverse memory buffer in reducing the gap between the empirical loss calculated over the buffer and the expected loss over the entire dataset. This is underpinned by the principle that a diverse buffer more accurately approximates the full data distribution, thereby improving the learning model's generalization capabilities.
>
> The apparent inconsistency between the theoretical use of covariance to measure diversity and the practical use of average distance among data points in the experiments arises from the need to adapt theoretical concepts for empirical validation. While covariance provides a measure of diversity that is theoretically sound, average distances serves as an effective and correlative approximation, which offer a more intuitive and directly observable metric that can be used in practical scenarios.
>
>
> > **Q1: Please explain more on the probability distance used in Section 3.1, e.g. the specific formulation.**
>
> We are sorry we did not make it clear.
>
> In equation 2, $ A(v, B_i) $ represents the distance measure between a node $ v $ and a set of nodes $ B_i $ in the buffer. The term 'L2 norm distance on probabilities' refers to the Euclidean distance calculated between the probability distributions of node features or embeddings. These probabilities are derived from the output of our model. Specifically, they represent the probability distribution of each node belonging to different classes, as obtained from the softmax output layer of the classifier.
>
> To clarify, for a given node $ v $ and the buffer $ B_i $, the L2 norm distance is calculated by $ A(v, B_i) = \min_{u \in B_i} \left( \sum_{j=1}^{K} (p^v_j - p^u_j)^2 \right) $, where $ p^v $ and $ p^u $ are the probability distribution of node $ v $ and $ u $, $ p^v_j $ and $ p^u_j $ are their $ j $-th item. $ K $ is the current class number.

---

> > ### Author Response · Authors · 2024-11-22
> >
> > > **Q2: Is the replay data generator trained from scratch each time when a new task comes in? Or is the generator train each time when a new task comes in, but is initialized from the generator from the previous task?**
> >
> > Yes, the generator train each time when a new task comes in, but is initialized from the generator from the previous task.
> >
> > The parameters of the variational layer are trained concurrently with other model parameters, with a focus on the data stored in the memory buffers. At each new task is introduced, the variational layer's parameters are updated, but crucially, this update incorporates all the previously accumulated data in the buffer. This update process allows the generative model to retain knowledge from earlier tasks, so the forgetting problem is prevented.
> >
> > > **Q3: If the generator is trained every time when a new task comes in, will this induce significant extra computational burden?**
> >
> > The generative memory replay to our model introduces new parameters solely from the variational layer. As depicted in Figure 2, this layer is comprised of MLP networks, which are inherently compact with a relatively small number of parameters.
> >
> > Importantly, the number of parameters in the variational layer remains constant, regardless of the arrival of new tasks. It learns the node distributions and is shared across different nodes. Thus, the parameters will not increase then the new node are introduced.

---

> ### Comment · Reviewer_Mh5c · 2024-11-24
>
> Thanks for the detailed responses from the authors. I guess that additional explanation on the theoretical analysis makes the paper more convincing. I would recommend that the authors to include the theoretical justification of 'sample data has >50% probability to underestimate the covariance' in the paper. But I would also like to say that >50% is actually not convincing enough. Because if it is only very slightly above 50% in practice, then it is no difference to 50%.  Since this is the major concern, I would like to increase  the rating to 6.

---

> > ### Author Response · Authors · 2024-11-24
> >
> > We appreciate your prompt feedback and increasing the score!  We will follow your instruction to include the theoretical justification regarding the probability of underestimating the covariance in the univariate case. Thank you once again for your constructive comments, which certainly help strengthen our paper.

---

### Official Review · Reviewer_8sJn · 2024-11-05

**Soundness:** 2
**Presentation:** 3
**Contribution:** 2
**Rating:** 5
**Confidence:** 4

**Summary:**

This paper presents a novel approach called Diversified Memory Selection and Generation (DMSG) for incremental learning in growing graphs. The authors address the challenge of continually training graph models to handle new tasks while retaining knowledge from previous tasks.

**Strengths:**

The developed model is supported by a theoretical foundation. The authors provide a theoretical analysis to support the importance of buffer diversity in incremental learning scenarios.

The studied problem is important. The authors try to tackle the issues of limited buffer size and potential overfitting through their diversified memory generation approach.

clear visualization and presentation. The authors use well-structured figures to convey the incremental learning process, memory selection, and generative replay methods in a relatively straightforward approach.

**Weaknesses:**

Lack of novelty. This paper largely builds on existing concepts of memory replay and buffer selection strategies that are already well-established in incremental learning and continual learning literature. While the paper proposes a diversified memory selection and generative replay approach, the techniques, such as adversarial learning and variational embedding, have been previously applied in other contexts. The paper simply combines different losses to facilitate learning.

Lack of baselines. The following papers should be discussed and compared for experiments:
Towards robust graph incremental learning on evolving graphs. ICML 2023
Replay-and-Forget-Free Graph Class-Incremental Learning: A Task Profiling and Prompting Approach. NeurIPS 2024
On the Limitation and Experience Replay for GNNs in Continual Learning. CoLLAs 2024
Topology-aware Embedding Memory for Continual Learning on Expanding Networks. KDD 2024
PUMA: Efficient Continual Graph Learning for Node Classification With Graph Condensation. TKDE 2024
Graph Continual Learning with Debiased Lossless Memory Replay. ECAI 2024
Accordingly, it is encouraged that the authors discuss the difference between their proposed model and these baselines.

Lack of comprehensive datasets. What about the performance of the proposed model on Citeseer, Pubmed, and PPI? How about Collab, IMDB, Proteins, and NCI1?

Lack of hyperparameter sensitivity study. The method introduces several hyperparameters such as λ1, λ2, and λ3 for balancing the loss. A discussion on their impact and guidelines for tuning these hyperparameters would be valuable. More in-depth analyses about balancing these losses to understand the contribution and reasons for developing them are encouraged.

**Questions:**

see above.

---

> ### Author Response · Authors · 2024-11-22
>
> > **W1: Lack of novelty. This paper largely builds on existing concepts of memory replay and buffer selection strategies that are already well-established in incremental learning and continual learning literature. While the paper proposes a diversified memory selection and generative replay approach, the techniques, such as adversarial learning and variational embedding, have been previously applied in other contexts. The paper simply combines different losses to facilitate learning.**
>
> We appreciate your feedback and the opportunity to further explain the novel contributions of our work. Our paper introduces the Holistic Diversified Memory Selection and Generation framework, specifically designed for incremental learning on growing graphs. This framework innovatively both proposed new algorithms and integrates several advanced techniques, tailored to overcome the unique challenges of graph-based continual learning. Below, we detail the distinctive aspects of each component:
>
> 1. **Greedy Algorithm for Diversified Memory Sampling with Theoretical Grounding**:
>
>     Our greedy algorithm for memory sampling is specifically designed to handle the complexity of graph structures, which is distinct from previous applications in general datasets. This algorithm considers both **intra-class and inter-class diversities**, which are critical in graph-based models due to their relational nature. We provide a rigorous **theoretical analysis** to support the effectiveness of our approach, proving that our diversified sampling strategy minimizes the loss in representational fidelity—a novel contribution that extends beyond typical greedy approaches in non-graph data.
>
> 2. **Adversarial Variational Generation-based Memory Replay**:
>
>     The use of adversarial variational techniques for memory replay in graphs is **an innovative approach within the domain of graph incremental learning**. This method not only aids in better generalization but also enhances the model’s ability to synthesize new graphs that maintain the integrity of past learnings—addressing the catastrophic forgetting problem more robustly than existing methods. This differs from traditional graph memory replay methods because it can introduce more diversified memories via variational generation.
>
> 3. **Holistic Diversified Memory Selection and Generation Framework**:
>
>     The holistic nature of our framework, which integrates both the diversified memory selection through a greedy algorithm and the generative replay via adversarial variational techniques, provides **a unique and comprehensive framework** that is specifically tailored for graph-based data. This is a key innovation over existing methods that often treat memory selection and replay as disjointed components.
>
> In conclusion, the proposed framework to the domain of incremental graph learning, coupled with theoretical underpinnings and empirical validations, contributes novel insights and practical method to the field.
>
>
> > **W2: Lack of baselines.  The following papers should be discussed and compared for experiments: Towards robust graph incremental learning on evolving graphs. ICML 2023 Replay-and-Forget-Free Graph Class-Incremental Learning: A Task Profiling and Prompting Approach. NeurIPS 2024 On the Limitation and Experience Replay for GNNs in Continual Learning. CoLLAs 2024 ... Accordingly, it is encouraged that the authors discuss the difference between their proposed model and these baselines.**
>
> Thank you for your comments and for pointing out relevant literature. We have checked the mentioned papers one by one carefully.
>
> The first three papers focus on task incremental learning, where the node's task affiliation is known. This is **different from** our setting—class-incremental learning—which does not presuppose knowledge of node class assignments in different tasks, presenting a more complex challenge.
>
> The last three papers were all published late this year, aligning with the **same period** as our paper. The fourth paper proposes parameter decoupled graph neural networks with topology-aware embedding memory for the graph incremental learning problem. The last two papers both use different graph condensation technology to preserve memory graphs. But still, they do not address memory diversification. Our method not only preserves but also enhances memory graphs through a learned generative model, fostering richer and more adaptable memory representations.  Furthermore, it is worth mentioning that our method employs a greedy algorithm for memory selection and an additional variational layer for memory selection. Compared to these papers, our require much fewer parameters.
>
> In response to these observations, we will include citations and a detailed discussion of these papers in the revised version of our manuscript.

---

> ### Author Response · Authors · 2024-11-22
>
> > **W3: Lack of comprehensive datasets. What about the performance of the proposed model on Citeseer, Pubmed, and PPI? How about Collab, IMDB, Proteins, and NCI1?**
>
> In the current study, we use the four datasets, CoraFull, OGB-Arxiv, Reddit, and OGB-Products, based on their common usage in the literature, which allows for direct comparison with state-of-the-art methods and ensures that our findings are relevant and comparable. Notably, most of the papers mentioned in the last question use these four datasets.
>
> Thank you for your suggestion and we will certainly consider including more datasets in subsequent studies based on your valuable feedback.
>
> > **W4:Lack of hyperparameter sensitivity study. The method introduces several hyperparameters such as λ1, λ2, and λ3 for balancing the loss. A discussion on their impact and guidelines for tuning these hyperparameters would be valuable. More in-depth analyses about balancing these losses to understand the contribution and reasons for developing them are encouraged.**
>
> We have conducted hyperparameter sensitivity experiments on several critical hyperparameters. For example, the effect of buffer size is analyzed in **Section A.5.3**.
> For the loss weights, we report the setting in our experiments in **A.4.3**.
> To tune these parameters, we employed a grid search strategy over a predefined range of values. The findings suggest that  $\lambda_2$ and  $\lambda_3$ are relatively robust across a wide range of settings, while $\lambda_1$ for the memory replay needs to be set as about 20 times higher weight than other weights. We analyze that there are two key reasons--Minimizing Forgetting: Since the model has already been trained on data in the buffers, the loss values for these classes are relatively smaller than for new classes. This discrepancy can lead the model to focus excessively on new classes, thereby exacerbating the forgetting problem. Addressing Imbalance Issues: The number of nodes in the buffers is significantly less than that of the new data, presenting an imbalance challenge. Increasing the weight helps address this by giving more importance to the smaller, buffered classes, which is a common approach in tackling imbalanced classification problems.

---

### Official Review · Reviewer_NW1X · 2024-11-10

**Soundness:** 3
**Presentation:** 2
**Contribution:** 3
**Rating:** 6
**Confidence:** 4

**Summary:**

This paper addresses the problem of incremental learning in growing graphs with new-coming node classes. It introduces a novel holistic Diversified Memory Selection and Generation (DMSG) framework containing two modules: (1) a buffer selection strategy that considers both intra-class and inter-class diversities, with a greedy algorithm; and (2) a diversified memory generation replay method with variational generation, with integrity loss and reconstructed loss under adversarial learning. Experiments over evolving graphs on node classification tasks could verify its effectiveness, along with ablation study over memory generation replay module.

**Strengths:**

S1. [Clear Motivation & Descriptions] The paper provides a well-defined motivation for addressing continual learning on growing graphs, focusing on memory selection and memory reply effectiveness. The reason for developing the proposed method can be compelling, along with clear and detailed descriptions of the problem background and definition, making it accessible and understandable.

S2: [Rationale Methodology] The rationale for the DMSG approach is well-grounded, combining heuristic buffer selection and generative memory reply to broaden the replay capability. The methodology introduces two strategies: first, a buffer selection algorithm that maximizes intra-class and inter-class diversity; second, a variational generation layer that synthesizes embeddings for effective replay, which sounds rational and novel to me.

S3. [Theoretical and Empirical Experiments] The paper validates its approach with both theoretical and empirical support, for instance, theoretical analysis to demonstrate the importance of effective buffer selection. Experimentally, the DMSG framework is tested on multiple evolving graph datasets, a good improvement over the AF/% metric.

**Weaknesses:**

Here are still some questions mainly regarding the buffer selection part and the variational part:

W1: For the buffer node selection part, are the buffer nodes only node-set or graph-structural data, if graph data, how to connect these nodes? and if node-set, how to directly feed it into GNN in Figure 2?  Further clarification on this would be appreciated.

W2: How many nodes are typically selected in the buffer for each class? How experimental results would change along with the varying number of nodes in buffers?

W3: In the ablation study, the effectiveness of the buffer node selection is not verified. For instance, how to illustrate the effectiveness of the proposed heuristic greedy method?

W4: Why the variational layer in the memory replays could broaden knowledge from the buffer? Does that mean, it generates something new $\hat{Z}$ from the original Z? Additionally, the minimization term in equation (7) is difficult to interpret—could you provide a more detailed explanation or expanded expressions for this term?

W5: With the introduction of complex mechanisms for memory selection and replay, how do the time and computational costs of the proposed method compare to other baseline approaches in the experiments?

W6: Why have a direct assumption that "Let the loss function L(θ, x) be β-Lipschitz continuous in respect to the input x" in Theorem 1. ? And how can we evaluate whether the selected nodes in buffers are with diversity?

**Questions:**

See Weakness.

---

> ### Author Response · Authors · 2024-11-21
>
> > **W1: For the buffer node selection part, are the buffer nodes only node-set or graph-structural data, if graph data, how to connect these nodes? and if node-set, how to directly feed it into GNN in Figure 2?**
>
> Thank you for the question. As mentioned in **Line 340-341**, we utilize a mini-batch GNN propagation approach. For each node, whether from the buffer or associated with new tasks, we extract its multi-hop neighborhood according to the original graph structure. Nodes are fed into the GNN along with their extracted neighborhood subgraphs. This aligns with practices in existing graph continual learning research.
>
> > **W2: How many nodes are typically selected in the buffer for each class? How experimental results would change along with the varying number of nodes in buffers?**
>
> Thank you for your inquiry. We report the number of nodes seleted for each datasets in **Section A.4.3**. In our experiments, we have standardized the buffer size to ensure consistency across evaluations and comparability with baseline methods. Specifically, we use a buffer size of 60 nodes for the CoraFull dataset and 400 nodes for the other datasets. This also aligns with the buffer sizes typically employed in baselines.
>
> To explore how the number of nodes in the buffer affects our model's performance, we conducted additional experiments with varying buffer sizes. The results are presented in **Figure 7 in Section A.5.3** of our manuscript. Our findings indicate that increasing the buffer size tends to improve the model’s ability to retain knowledge, particularly for more complex datasets, but up to a point where the returns diminish. This helps in understanding the trade-offs involved in buffer size selection concerning memory constraints and performance optimization.
>
> > **W3: In the ablation study, the effectiveness of the buffer node selection is not verified. For instance, how to illustrate the effectiveness of the proposed heuristic greedy method?**
>
> We have conducted experiments to analyze the effectiveness of the proposed heuristic greedy method for buffer node selection. The results are in **Table 4 in Section A.5.2**, we compare our selection method with other widely used memory selection methods, the results show that our sampling method consistently excels in performance. This can be
> attributed to exploring both the intra-class and inter-class diversity of nodes within our method that efficiently captures the required diversity among nodes, ensuring a holistic representation of the dataset.
>
> > **W4: Why the variational layer in the memory replays could broaden knowledge from the buffer? Does that mean, it generates something new from the original Z? Additionally, the minimization term in equation 7 is difficult to interpret—could you provide a more detailed explanation or expanded expressions for this term?**
>
> Thank you for your insightful questions.  The variational layer in our model is designed to enhance the diversity and generalization capabilities of the buffer memory by generating new node embeddings from the existing data (original $Z$). This is achieved through a process known as variational inference, where the layer learns to approximate the probability distribution of the node embeddings based on the input data. Yes, the variational layer effectively generates new samples from the learned distribution, which are similar but not identical to the original data $Z$. These synthesized embeddings are derived by sampling from a distribution parameterized by the means $Z^{\mu}$ and variances $Z^{\sigma}$ produced by the variational layer. This capability allows the model to interpolate and extrapolate within the space of known data, thereby enhancing the model's ability to handle new or slightly different data scenarios without needing explicit examples of such variations.
>
> Equation 7 employs a min-max optimization strategy within an adversarial framework. The term is expressed as:
>
> $\min_{\theta,\phi_v}\max_{\phi_d}E_{z_i \sim Z}E_{q_{\phi_v}(\hat{z}_i|z)} [l_D(z_i,\hat{z}_i)] $
>
> which can also be written as a bilevel optimization expression:
>
> $ \min_{\theta, \phi_v} \quad E_{z_i \sim Z}E_{q_{\phi_v}(\hat{z}_i|z)} [l_D(z_i, \hat{z}_i, \phi_d^* )]$
>
> subject to
>
> $ \phi_d^* = \arg\max_{\phi_d} \quad E_{z_i \sim Z}E_{q_{\phi_v}(\hat{z}_i|z)} [l_D(z_i, \hat{z}_i)]$
>
> Here, $\theta$ and $\phi_v$ represent the parameters of the main network and the variational network, respectively, and $\phi_d$ represents the parameters of a discriminator network. The $q_{\phi_v}(\hat{z}_i|z)$ is the variational distribution to sample $\hat{z}_i$. The discriminator $D$ attempts to distinguish between original and generated node embeddings, whereas the generator (part of the variational layer) tries to fool the discriminator by producing embeddings that are indistinguishable from actual data. The loss $l_D$ is a binary cross-entropy loss, measuring how well the discriminator performs and how effectively the generator deceives it.

---

> ### Author Response · Authors · 2024-11-21
>
> > **W5: With the introduction of complex mechanisms for memory selection and replay, how do the time and computational costs of the proposed method compare to other baseline approaches in the experiments?**
>
> The main part of computational costs is the memory selection strategy, and the memory replay stage involves similar quantities of propagation costs across all baselines.
>
> We analyze the time complexity of our method in **Line 240-244 in Section 3.1**, which demonstrates that our method is efficient and comparable with other straightforward strategies such as K-center, Covering Machines (CM), and Matrix Factorization (MF). And our method consistently excels in performance.
>
> For a more detailed time complexity analysis for these memory selection methods, please refer to the response to **Review T2ad in W3**. where we have further detailed time complexity.
>
> > **W6: Why have a direct assumption that "Let the loss function L(θ, x) be β-Lipschitz continuous in respect to the input x" in Theorem 1. ? And how can we evaluate whether the selected nodes in buffers are with diversity?**
>
> The assumption that the loss function $L(\theta,x)$ is β-Lipschitz continuous with respect to the input $x$ is a standard theoretical simplification that ensures the stability and smoothness of the loss function. It is widely used as an established assumption for graph neural networks[1].
>
> We evaluate the diversity of the selected nodes using both intra-class and inter-class diversity metrics. Intra-class diversity ensures that the nodes within the same class in the buffer are varied, covering different aspects or features of the class. Inter-class diversity, on the other hand, ensures that nodes from different classes are distinct enough to reinforce class boundaries effectively. Specific metrics might include variance measures within classes for intra-class diversity and distance metrics like Euclidean or cosine distances for inter-class diversity. In this paper, we use the Euclidean distance.
>
> [1] Levie, Ron. "A graphon-signal analysis of graph neural networks." Advances in Neural Information Processing Systems 36 2024.

---

### Comment · Area_Chair_5afA · 2024-11-25
**Please reply to the authors' response.**

Dear reviewers,

The ICLR author discussion phase is ending soon. Could you please review the authors' responses and take the necessary actions? Feel free to ask additional questions during the discussion. If the authors address your concerns, kindly acknowledge their response and update your assessment as appropriate.


Best,
AC

---

### Meta-Review · Area_Chair_5afA · 2024-12-17

**Metareview:**

The paper addresses the challenge of incremental learning in growing graphs, aiming to retain inference ability on previous tasks while learning new ones.

The reviewers highlighted several strengths of the paper, including its clear motivation, innovative methodology, and detailed theoretical analysis. The experimental results were praised for their robustness and clarity. However, some concerns were raised regarding the buffer node selection process, the role of the variational layer, and the time complexity of the proposed methods. The authors addressed these concerns effectively, providing additional justifications and detailed analyses.

Given the overall positive reception, I recommend its acceptance.

**Additional Comments On Reviewer Discussion:**

The authors addressed most concerns effectively, providing additional justifications and detailed analyses. Two reviewers acknowledged that their concerns were addressed (one reviewer increased the rating).

---

### Decision · Program_Chairs · 2025-01-22

Accept (Poster)